# CHAIN-OF-TABLE: EVOLVING TABLES IN THE REASONING CHAIN FOR TABLE UNDERSTANDING

**Zilong Wang**[1][*]   **Hao Zhang**[3]   **Chun-Liang Li**[2]   **Julian Martin Eisenschlos**[3]
**Vincent Perot**[3]   **Zifeng Wang**[2]   **Lesly Miculicich**[2]   **Yasuhisa Fujii**[3]
**Jingbo Shang**[1]   **Chen-Yu Lee**[2]   **Tomas Pfister**[2]
[1]University of California, San Diego   [2]Google Cloud AI Research   [3]Google Research

## ABSTRACT

Table-based reasoning with large language models (LLMs) is a promising direction to tackle many table understanding tasks, such as table-based question answering and fact verification. Compared with generic reasoning, table-based reasoning requires the extraction of underlying semantics from both free-form questions and semi-structured tabular data. Chain-of-Thought and its similar approaches incorporate the reasoning chain in the form of textual context, but it is still an open question how to effectively leverage tabular data in the reasoning chain. We propose the CHAIN-OF-TABLE framework, where tabular data is explicitly used in the reasoning chain as a proxy for intermediate thoughts. Specifically, we guide LLMs using in-context learning to iteratively generate operations and update the table to represent a tabular reasoning chain. LLMs can therefore *dynamically plan* the next operation based on the results of the previous ones. This continuous evolution of the table forms a chain, showing the reasoning process for a given tabular problem. The chain carries structured information of the intermediate results, enabling more accurate and reliable predictions. CHAIN-OF-TABLE achieves new state-of-the-art performance on WikiTQ, FeTaQA, and TabFact benchmarks across multiple LLM choices.

## 1 INTRODUCTION

Tables are a popular data format and widely used in daily life (Cafarella et al., 2008). Understanding tabular data with language models can benefit various downstream tasks, such as table-based fact verification (Chen et al., 2019), and table-based question answering (Jin et al., 2022). Distinct from pure text, tables deliver rich information through the interaction between rows and columns in the tabular structure, which enhances the data capacity but also increases the difficulty for language models to understand them. Thus, reasoning over the tabular data is an important direction in natural language processing and attracts increasing attention from both academia and industry.

In recent years, several approaches have been suggested to tackle the problem of table understanding by *training* language models. One common direction is to add specialized embedding layers or attention mechanisms into language models and pre-train the models by recovering table cells or segments (Herzig et al., 2020; Wang et al., 2021; Gu et al., 2022; Andrejczuk et al., 2022). In this way, the pre-trained models are aware of the tabular structure. Another direction is to synthesize SQL query-response pairs and pre-train an encoder-decoder model as a neural SQL executor (Eisenschlos et al., 2020; Liu et al., 2021; Jiang et al., 2022).

Recently, large language models (LLMs) achieve outstanding performance across diverse tasks solely by *prompting*, thanks to the massive scale of pre-training (Brown et al., 2020; Kojima et al., 2022). As series of works on prompting techniques have further improved the reliability of LLMs by designing reasoning chains, such as Chain-of-Thought (Wei et al., 2022), Least-to-Most (Zhou et al., 2022), Program-of-Thought (Chen et al., 2022) and Tree-of-Thought (Yao et al., 2023). Different works have also explored the possibility of using LLMs to solve table-based problems (Chen,

---

[*]Work done while the author was a student researcher at Google Cloud AI Research. Correspondence to: Zilong Wang <zlwang@ucsd.edu>, Chen-Yu Lee <chenyulee@google.com>.

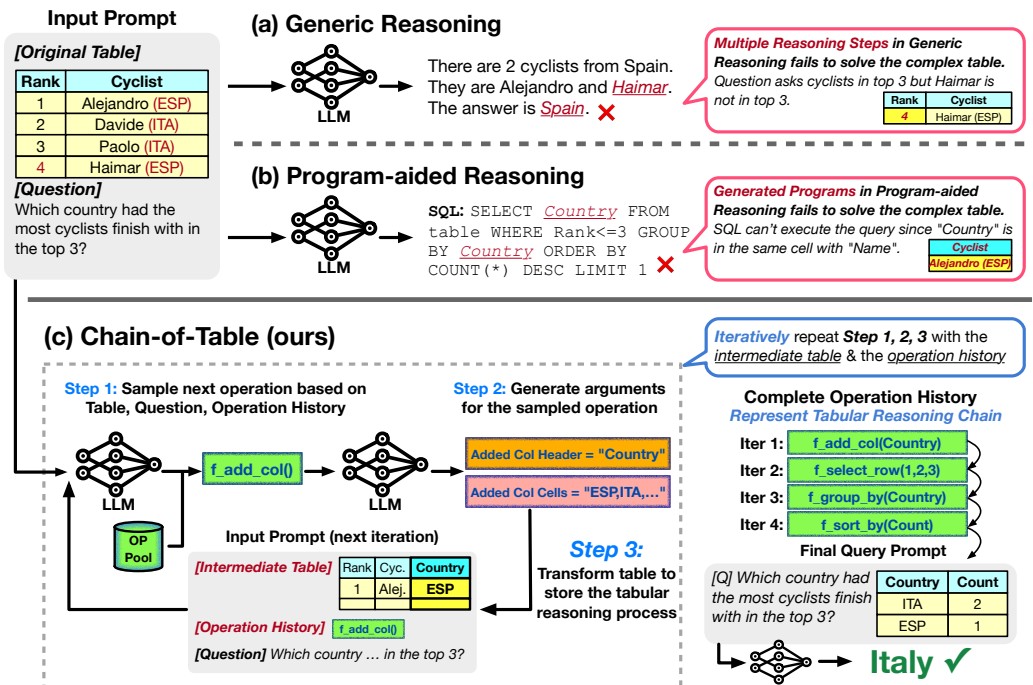

Figure 1: Illustration of the comparison between (a) generic reasoning, (b) program-aided reasoning, and (c) the proposed CHAIN-OF-TABLE. Given a complex table where a cyclist's nationality and name are in the same cell, (a) is unable to provide the correct answer through multi-step reasoning due to the complexity; (b) generates and executes programs (e.g. SQL queries) to deliver the answer, but it also falls short in accurately parsing the name and nationality in the table. In contrast, (c) CHAIN-OF-TABLE iteratively samples a chain of operations that effectively transform the complex table into a version specifically tailored to the question. With the assistance of CHAIN-OF-TABLE, the LLM can arrive at the correct answer.

2023; Cheng et al., 2022; Ye et al., 2023). However, these approaches (Hsieh et al., 2023) often represent reasoning steps in free-form text or code, which are not ideally suited for addressing scenarios involving complex tables, as shown in Figure 1(a) and Figure 1(b).

On the other hand, inference on tables typically involves a series of intermediate reasoning steps and each of them aligns with specific tabular operations. We propose CHAIN-OF-TABLE, where we conduct step-by-step reasoning as step-by-step tabular operations to form a *chain* of tables. The tables in the chain are the transformed tables by the tabular operations, representing the intermediate reasoning results. This procedure resembles the *thought* of reasoning in Chain-of-Thought (Wei et al., 2022). Specifically, we define a set of table operations, such as adding columns, selecting rows, grouping, and more, which are commonly-used in SQL and DataFrame development (Pönighaus, 1995; Shi et al., 2020; Katsogiannis-Meimarakis & Koutrika, 2023). We then prompt LLMs to conduct step-by-step reasoning. In each step, the LLM dynamically generates an operation as the next step along with its required arguments, and then we execute the operation on the table programmatically. This operation can either enrich the table by adding detailed intermediate results or condense it by removing irrelevant information. Intuitively, visualizing the intermediate results is essential for reaching correct predictions. We feed the transformed table back for the next step. This iterative process continues until an ending state is achieved. We argue that the tables obtained during the reasoning steps are better structured representations of the intermediate thoughts than free-form text. Finally, the CHAIN-OF-TABLE reasoning results in tables from which it is easier for LLMs to derive a final answer to the question.

We validate CHAIN-OF-TABLE with three tabular benchmarks to evaluate table-based reasoning: WikiTQ (Pasupat & Liang, 2015), TabFact (Chen et al., 2019), and FeTaQA (Nan et al., 2022). We conduct our experiments using PaLM 2 (Anil et al., 2023) and GPT-3.5 (Brown et al., 2020; OpenAI,

2023) to demonstrate that our proposed method CHAIN-OF-TABLE is able to generalize to various LLM options. We summarize our contribution as follows:

- We extend the concept of Chain-of-Thought to the tabular setting, where we transform the input table to store intermediate results. This multi-step tabular reasoning approach with table evolution leads to more accurate table understanding.
- Extensive experiments on table-based fact verification and question answering show that CHAIN-OF-TABLE archives state-of-the-art performance in WikiTQ, TabFact, and FeTaQA datasets.

## 2   RELATED WORK

**Fine-tuning Language Model for Table Understanding**   Tables are effective in organizing, storing, and analyzing information. Efforts have been made to fine-tune language models (LMs) to tackle table understanding tasks. Following the successful mask language modeling (MLM) proposed in BERT (Devlin et al., 2019), TaPas (Herzig et al., 2020) adopts this approach and asks the model to reconstruct certain cells in the table during pre-training. Pasta (Gu et al., 2022) and TUTA (Wang et al., 2021) further propose to mask the entire columns or segments in the table. On the other hand, TAPEX (Liu et al., 2021) pre-trains an encoder-decoder model with a large synthetic SQL dataset so that it can perform as a SQL executor to better understand the tabular structure. Eisenschlos et al. (2020) and Jiang et al. (2022) also leverage synthesized SQL with additional consideration of the alignment between SQL and natural language questions by pre-training the model with both natural and synthetic data.

**Prompting Language Model for Table Understanding**   LLMs can learn from a few samples as prompts through in-context learning. This strategy is widely used to give models additional instructions to better solve downstream tasks. Chain-of-Thought (CoT) (Wei et al., 2022) proposes to generate reasoning steps before answering instead of directly generating an end-to-end answer. Following CoT, Least-to-Most (Zhou et al., 2022) and DecomP (Khot et al., 2022) propose to break down the question into subproblems in the reasoning chain. During reasoning, the latter steps are aware of the previous ones. Such iterative chains with task decomposition further improve the results on complex problems by leveraging the intermediate results from solving subproblems. Jin & Lu (2023) enhances CoT through a table-filling procedure, with a primary focus on text-based tasks where the input and output are in textual format. However, the line of works following CoT is not specifically designed for tabular data. As reported in Chen (2023), large language models with these generic reasoning methods can achieve decent results, but there are still gaps between these methods and those specialized for table scenarios (Cheng et al., 2022; Ye et al., 2023). We propose CHAIN-OF-TABLE to fill the gap by directly incorporating intermediate tables from tabular operations as a proxy of intermediate thoughts.

To better solve table-based tasks with LLMs, researchers go beyond general text and resort to using external tools. Chen et al. (2022); Gao et al. (2023) propose solving reasoning tasks by generating Python programs, which are then executed using the Python interpreter. This approach greatly improves the performance of arithmetic reasoning. In the scenario of table understanding, Text-to-SQL with LLMs (Rajkumar et al., 2022) is a straightforward application of this idea. To further push the limits of programs, Binder (Cheng et al., 2022) generates SQL or Python programs and extends their capabilities by calling LLMs as APIs in the programs. LEVER (Ni et al., 2023) also proposes solving the table-based tasks with programs but with the additional step of verifying the generated programs with their execution results. However, the assistant programs in these program-aided methods still fall short in solving difficult cases that involve complex tables. These limitations are primarily due to the constraints of the *single-pass* generation process, where the LLMs lack the capability to modify the table in response to a specific question, requiring them to perform reasoning over a static table. Our method, on the contrary, is a *multi-step* reasoning framework that conducts tabular reasoning step by step. It transforms the tables tailored to the given question.

To the best of our knowledge, Dater (Ye et al., 2023) is the only model that modifies the tabular context while solving table-based tasks. However, the table decomposition in Dater is motivated by the idea that tables could be too large for LLMs to conduct reasoning. It is, therefore, more similar to an LLM-aided data pre-processing than to a part of the reasoning chain since the tabular operations are limited to column and row selections, and fixed for all tables and questions. In contrast, our

CHAIN-OF-TABLE generalizes a larger set of generic table operations and *dynamically* generates reasoning chains in an adaptive way based on the inputs, leveraging the planning ability (Valmeekam et al., 2022; Hao et al., 2023) of LLMs.

## 3   CHAIN-OF-TABLE REASONING

**Problem Formulation.**   In table-based reasoning, each entry can be represented as a triplet $(T, Q, A)$, where $T$ stands for the table, $Q$ represents a question or statement related to the table, and $A$ is the expected answer. Particularly, in the table-based question answering task, $Q$ and $A$ are the question and expected answer in natural language form; in the table-based fact verification task, $Q$ is a statement about the table contents and $A \in \{\text{True}, \text{False}\}$ is a Boolean value that indicates the statement's correctness. The objective is to predict the answer $A$ given the question $Q$ and the table $T$. To facilitate table-based reasoning within the same paradigm employed for generic reasoning, we convert all data values, including tables, into textual representations (see Appendix D for the tabular format encoding method).

### 3.1   OVERVIEW

CHAIN-OF-TABLE enables LLMs to *dynamically plan* a chain of operations over a table $T$ in response to a given question $Q$. It utilizes atomic tool-based operations to construct the table chain. These operations include adding columns, selecting rows or columns, grouping, and sorting, which are common in SQL and DataFrame development (see Appendix A for more details).

Previously, Dater (Ye et al., 2023) employs a dedicated yet fixed procedure for decomposing tables and questions, which limits its compatibility with new operations. Also, Binder (Cheng et al., 2022), while potentially compatible with new operations, is restricted to those that work with code interpreters such as SQL or Python. In contrast, our framework is extendable and can incorporate operations from a wide range of tools thanks to the flexible in-context learning capability to sample and execute effective operations.

As illustrated in Algorithm 1, at each iteration, we prompt the LLM to sample one of the pre-defined atomic operations denoted as f using the corresponding question $Q$, the latest table state $T$, and the operation chain chain (Line 4). Then, we query the LLM to generate the required arguments args for f (Line 5) and execute it to transform the table $T$ (Line 6). We keep track of the operation f performed on the table in the operation chain chain (Line 7). The process finishes when the ending tag [E] is generated (Line 8). Finally, we feed the latest table into the LLM to predict the answer (Line 9). This series of operations serves as the reasoning steps leading LLMs to understand the input table and better generate the final answer.

---

**Algorithm 1:** CHAIN-OF-TABLE Prompting

**Data:** $(T, Q)$ is a table-question pair.
**Result:** $\hat{A}$ is the predicted answer to the question.

1 **Function** Chain-of-Table $(T, Q)$:
2     chain $\leftarrow$ [([B],$\phi$),]             ▷ Initialize the operation chain chain with [B] and $\phi$, where [B] is
                                                       ▷ the beginning tag, and $\phi$ means it requires no arguments
3     **repeat**
4        f $\leftarrow$ DynamicPlan($T$,$Q$,chain)       ▷ Generate next operation f based on the table, the question, and
                                                          ▷ the current operation chain
5        args $\leftarrow$ GenerateArgs($T$,$Q$,f)          ▷ Generate the arguments args for the next operation
6        $T \leftarrow$ f($T$,args)                  ▷ Perform the next operation on the table to obtain updated $T$
7        chain $\leftarrow$ chain.$append$((f,args))       ▷ Keep track of the operations in the operation chain chain
8     **until** f $=$ [E]                ▷ Iteratively update the table until the ending tag [E] is generated
9     $\hat{A} \leftarrow$ Query($T$,$Q$)             ▷ Query the LLM with the resulting table to get the final answer $\hat{A}$
10  **return** $\hat{A}$

---

### 3.2   DYNAMIC PLANNING

CHAIN-OF-TABLE instructs the LLM to dynamically plan the next operation by in-context learning. As shown in Figure 2(a), DynamicPlan involves three components: the most recent intermediate table $T$ (Figure 2(a)(i)), the history of the previous operations chain chain (Figure 2(a)(ii)), and

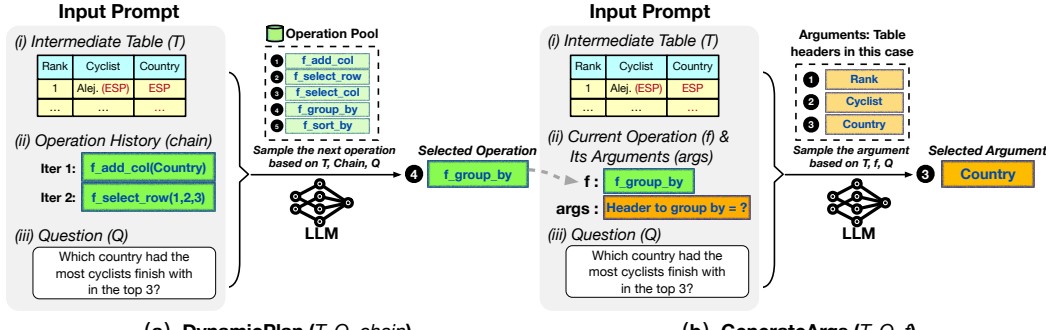

Figure 2: Illustration of `DynamicPlan`($T$,$Q$,`chain`) and `GenerateArgs`($T$,$Q$,`f`) in the proposed CHAIN-OF-TABLE, where $T$ is a intermediate table; $Q$ is the question; `chain` is a list of operations already performed on the table; `f` is the operation selected by `DynamicPlan`. **Left:** `DynamicPlan` samples the next operation from the operation pool, according to ($T$, `chain`, $Q$). **Right:** `GenerateArgs` takes the selected operation `f` as input and generates its arguments based on ($T$, `f`, $Q$). The operations, along with their arguments, act as a proxy of the tabular reasoning process to effectively tackle table understanding tasks.

the question $Q$ (Figure 2(a)(iii)). We guide the LLM to select the subsequent operation `f` from the operation pool given ($T$, `chain`, $Q$). The LLM is then able to dynamically plan the next operation and build a tabular reasoning chain step by step. See Appendix E.1 for detailed prompts.

### 3.3 ARGUMENT GENERATION

The next step, `GenerateArgs`, involves generating arguments for the selected table operation `f` sampled by `DynamicPlan`, as depicted in Figure 2. `GenerateArgs` involves three key components: the most recent intermediate table $T$ (Figure 2(b)(i)), the selected operation `f` along with its arguments `args` (Figure 2(b)(ii)), and the question (Figure 2(b)(iii)). We employ simple regular expressions to account for varying number of arguments required by different operations (see Appendix E.2 for more details). Finally, we apply programming languages to execute the operation and create the corresponding intermediate tables.

### 3.4 FINAL QUERY

We transform the table through dynamic planning (Section 3.2) and argument generation (Section 3.3). During this process, we create a chain of operations that acts as a proxy for the tabular reasoning steps. These operations generate intermediate tables that store and present the results of each step to the LLM. Consequently, the output table from this chain of operations contains comprehensive information about the intermediate phases of tabular reasoning. We then employ this output table in formulating the final query. As illustrated in Figure 1 (bottom right), we input both the output table and the question into the LLM, which provides the final answer to the question (see Line 9 in Algorithm 1).

## 4 EXPERIMENTS

We evaluate the proposed CHAIN-OF-TABLE on three public table understanding benchmarks: Wik-iTQ (Pasupat & Liang, 2015), FeTaQA (Nan et al., 2022), and TabFact (Chen et al., 2019). WikiTQ and FeTaQA are datasets focused on table-based question answering. They require complex tabular reasoning over the provided table to answer questions. WikiTQ typically requires short text span answers, whereas FeTaQA demands longer, free-form responses. TabFact, on the other hand, is a table-based binary fact verification benchmark. The task is to ascertain the truthfulness of a given statement based on the table. For WikiTQ evaluation, we use the official denotation accuracy (Pasupat & Liang, 2015), and for TabFact, we employ the binary classification accuracy. Given the nature of FeTaQA, which involves comparing predictions with longer target texts, we utilize BLEU (Papineni et al., 2002), ROUGE-1, ROUGE-2, and ROUGE-L (Lin, 2004) for assessment. In

Table 1: Table understanding results on WikiTQ and TabFact with PaLM 2 and GPT 3.5. (underline denotes the second-best performance; **bold** denotes the best performance; the improvement is measured against the second-best performing method.)

| Prompting | PaLM 2 | | GPT 3.5 | |
|---|---|---|---|---|
| | TabFact | WikiTQ | TabFact | WikiTQ |
| *Generic Reasoning* | | | | |
| End-to-End QA | 77.92 | 60.59 | 70.45 | 51.84 |
| Few-Shot QA | 78.06 | 60.33 | 71.54 | 52.56 |
| Chain-of-Thought (Wei et al., 2022) | 79.05 | 60.43 | 65.37 | 53.48 |
| *Program-aided Reasoning* | | | | |
| Text-to-SQL (Rajkumar et al., 2022) | 68.37 | 52.42 | 64.71 | 52.90 |
| Binder (Cheng et al., 2022) | 76.98 | 54.88 | 79.17 | 56.74 |
| Dater (Ye et al., 2023) | 84.63 | 61.48 | 78.01 | 52.81 |
| CHAIN-OF-TABLE (ours) | **86.61** (+1.98) | **67.31** (+5.83) | **80.20** (+1.03) | **59.94** (+3.20) |

our experiments, we use PaLM 2-S[1], GPT 3.5 (turbo-16k-0613)[2] as the backbone LLMs. We incorporate few-shot demo samples from the training set into the prompts to perform in-context learning. Examples of these prompts can be found in Appendix E. Details regarding the LLM inference parameters and the number of demonstration samples used are provided in Appendix C.

## 4.1 BASELINES

The baseline methods are categorized into two groups: (a) generic reasoning, which includes End-to-End QA, Few-Shot QA, Chain-of-Thought (Wei et al., 2022); and (b) program-aided reasoning, which includes Text-to-SQL (Rajkumar et al., 2022), Binder (Cheng et al., 2022), Dater (Ye et al., 2023)). Detailed descriptions of these baseline methods are provided below.

**Generic Reasoning**   End-to-End QA guides the LLM to directly produce the answer when provided with a table and a question as input prompts. Few-Shot QA operates similarly, but it includes few-shot examples of (Table, Question, Answer) triplets in the prompt, as detailed in Brown et al. (2020). We select these examples from the training set, and the model also outputs the answer directly. Chain-of-Thought (Wei et al., 2022) prompts the LLM to articulate its reasoning process in text format before delivering the question. See Appendix F for the prompts of baselines.

**Program-aided Reasoning**   Text-to-SQL (Rajkumar et al., 2022) utilizes in-context samples to guide LLMs in generating SQL queries for answering questions. This approach follows the concepts introduced by Chen et al. (2022); Gao et al. (2023). Binder (Cheng et al., 2022) integrates a language model API with programming languages such as SQL or Python. This integration prompts the LLM to produce executable programs that perform table reasoning tasks on the given table and question. Dater (Ye et al., 2023) employs few-shot samples for efficient deconstruction of table contexts and questions, enhancing end-to-end table reasoning with decomposed sub-tables and sub-questions.

## 4.2 RESULTS

We compare CHAIN-OF-TABLE with generic reasoning methods and program-aided reasoning methods on three datasets: WikiTQ, TabFact, and FeTaQA. The results on WikiTQ and TabFact are presented in Table 1. We have additional results on FeTaQA in Appendix B. We follow the previous works and report the performance using the official evaluation pipeline[3].

---

[1] https://cloud.google.com/vertex-ai/docs/generative-ai/learn/generative-ai-studio
[2] http://openai.com/api/
[3] Dater Ye et al. (2023) with OpenAI Codex LLM achieves 65.9% and 85.6% accuracy on WikiTQ and TabFact, respectively. It also achieves 27.96 in BLEU, 0.62 in ROUGE-1, 0.40 in ROUGE-2, and 0.52 in ROUGE-L on FeTaQA. However, because Codex is no longer publicly available, we do not compare CHAIN-OF-TABLE with Dater with Codex.

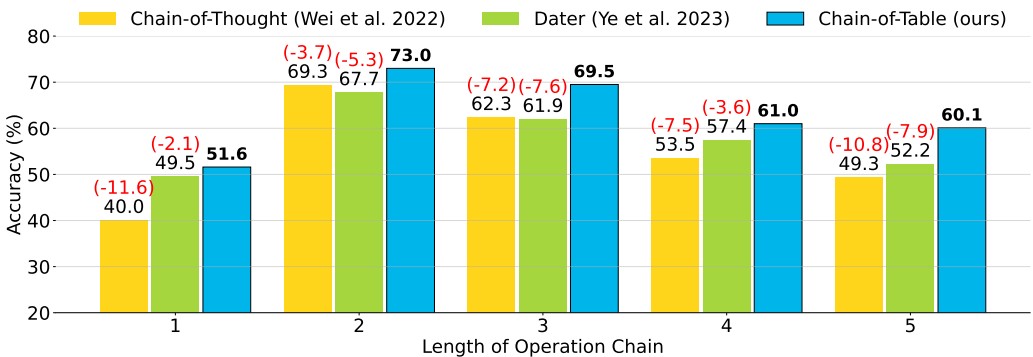

Figure 3: Performance of Chain-of-Thought, Dater, and the proposed CHAIN-OF-TABLE on Wik-iTQ for questions that require an operation chain of varying lengths. Our proposed atomic operations allow our proposed method CHAIN-OF-TABLE to dynamically transform the input table through multiple reasoning iterations. This significantly improves performance over generic and program-aided reasoning counterparts.

Table 2: Distribution of the number of samples v.s. the required length of operation chain in CHAIN-OF-TABLE with PaLM 2 on WikiTQ and TabFact datasets. We observe that the majority of samples need 2 to 4 operations to generate the final output.

| Dataset | Length of operation chain | | | | |
|---------|----|------|------|------|-----|
|         | 1  | 2    | 3    | 4    | 5   |
| WikiTQ  | 95 | 1308 | 1481 | 1084 | 341 |
| TabFact | 4  | 547  | 732  | 517  | 223 |

Table 1 shows that CHAIN-OF-TABLE significantly outperforms all generic reasoning methods and program-aided reasoning methods on TabFact and WikiTQ across PaLM 2 and GPT 3.5. This is attributed to the dynamically sampled operations and the informative intermediate tables in CHAIN-OF-TABLE. CHAIN-OF-TABLE iteratively generates operations that act as proxies for tabular reasoning steps. These operations produce and present tailored intermediate tables to the LLM, conveying essential intermediate thoughts (see the example in Figure 4). With the support of CHAIN-OF-TABLE, the LLM can reliably reach the correct answer.

From the results, we observe a performance decrease on WikiTQ due to the complexity of tabular structure when vanilla Chain-of-Thought is introduced to End-to-End QA using PaLM 2. In contrast, our proposed CHAIN-OF-TABLE consistently enhances End-to-End QA performance by 8.69% on TabFact and 6.72% on WikiTQ with PaLM 2.

## 4.3 PERFORMANCE ANALYSIS UNDER DIFFERENT OPERATION CHAIN LENGTHS

In CHAIN-OF-TABLE, the selection of each operation is dynamically determined based on the difficulty and complexity of the questions and their corresponding tables. Therefore, we conduct a detailed study on the performance under different numbers of operations by categorizing the test samples according to their operation lengths. We report the distribution of the number of samples v.s. the required length of operation chain in Table 2. This analysis focuses on samples that require operations in the reasoning process. We use the results with PaLM 2 as an example. Our observations reveal that the majority of samples require 2 to 4 operations to generate the final output.

For each chain length, we further compare CHAIN-OF-TABLE with Chain-of-Thought and Dater, as representative generic and program-aided reasoning methods, respectively. We illustrate this using results from PaLM 2 on WikiTQ. We plot the accuracy of all methods using bar charts in Figure 3, highlighting the gap between the compared methods and our method. Notably, CHAIN-OF-TABLE consistently surpasses both baseline methods across all operation chain lengths, with a significant margin up to 11.6% compared with Chain-of-Thought, and up to 7.9% compared with Dater.

Table 3: Performance of Binder, Dater, and the proposed CHAIN-OF-TABLE on small (<2000 tokens), medium (2000 to 4000 tokens), large (>4000 tokens) tables from WikiTQ. We observe that the performance decreases with larger input tables while CHAIN-OF-TABLE diminishes gracefully, achieving significant improvements over competing methods. (underline denotes the second-best performance; **bold** denotes the best performance; the improvement is measured against the second-best performing method.)

| Prompting | Table Size | | |
|---|---|---|---|
| | Small (<2k) | Medium (2k~4k) | Large (>4k) |
| Binder (Cheng et al., 2022) | 56.54 | 26.13 | 6.41 |
| Dater (Ye et al., 2023) | 62.50 | 42.34 | 34.62 |
| CHAIN-OF-TABLE (ours) | **68.13** (+5.63) | **52.25** (+9.91) | **44.87** (+10.25) |

Table 4: Number of samples generated for a single question in Binder, Dater, and the proposed CHAIN-OF-TABLE on the WikiTQ dataset. Notably, CHAIN-OF-TABLE generates the fewest samples among the baselines – 50% less than Binder and 75% less than Dater. For a detailed description of the steps involved in Binder and Dater, please refer to the corresponding papers.

| Prompting | Total # of generated samples | # of generated samples in each steps |
|---|---|---|
| Binder (Cheng et al., 2022) | 50 | Generate Neural-SQL: 50 |
| Dater (Ye et al., 2023) | 100 | Decompose Table: 40; Generate Cloze: 20; Generate SQL: 20; Query: 20 |
| CHAIN-OF-TABLE (ours) | ≤**25** | DynamicPlan: ≤5; GenerateArgs: ≤19; Query: 1 |

Generally, the performance of these methods decreases as the number of tabular operations required in the tabular reasoning chain increases due to higher difficulty and complexity of questions and tables. Nevertheless, our proposed CHAIN-OF-TABLE declines gracefully compared to other baseline methods. For example, CHAIN-OF-TABLE exhibits only a minimal decrease in performance when the number of operations increases from four to five.

## 4.4 PERFORMANCE ANALYSIS UNDER DIFFERENT TABLE SIZES

Large tables present significant challenges to LLMs since LLMs often struggle to interpret and integrate contexts in long input prompts (Liu et al., 2023a; Ye et al., 2023). To assess the performance on tables of various sizes, we categorize the input tables from WikiTQ into 3 groups based on token count: small (<2000 tokens), medium (2000 to 4000 tokens) and large (>4000 tokens). We then compare CHAIN-OF-TABLE with Dater (Ye et al., 2023) and Binder (Cheng et al., 2022), the two latest and strongest baselines, as representative methods. Detailed results are presented in Table 3.

As anticipated, the performance decreases with larger input tables, as models are required to process and reason through longer contexts. Nevertheless, the performance of the proposed CHAIN-OF-TABLE diminishes gracefully, achieving a significant 10+% improvement over the second best competing method when dealing with large tables. This demonstrates the efficacy of the reasoning chain in handling long tabular inputs.

## 4.5 EFFICIENCY ANALYSIS OF CHAIN-OF-TABLE

We analyze the efficiency of CHAIN-OF-TABLE by evaluating the number of required generated samples. We compare CHAIN-OF-TABLE with Binder (Cheng et al., 2022) and Dater (Ye et al., 2023), the two latest and most competitive baseline method. The analysis results on WikiTQ are presented in Table 4. Binder generates Neural-SQL queries, requiring 50 samples for self-consistent results. Dater involves multiple delicate yet fixed steps, such as decomposing the tables and generating cloze queries for the questions. In each step, Dater also employs self-consistency to improve accuracy of the LLM outputs, leading to a high number of required generated samples. For a de-

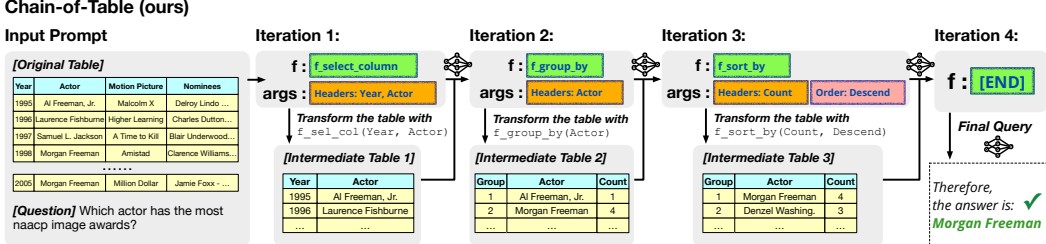

Figure 4: Illustration of the tabular reasoning process in CHAIN-OF-TABLE. This iterative process involves dynamically planning an operation chain and accurately storing intermediate results in the transformed tables. These intermediate tables serve as tabular thought process that can guide the LLM to land to the correct answer more reliably.

tailed description of these frameworks, please refer to the corresponding papers, Ye et al. (2023) and Cheng et al. (2022).

Unlike these previous methods, our proposed CHAIN-OF-TABLE employs a greedy search strategy in its tabular reasoning process, instead of relying on self-consistency sampling for boosting performance. This approach results in a reduced query count for our method, despite CHAIN-OF-TABLE adopting an iterative reasoning process. To be more specific, we observe that the number of queries needed by CHAIN-OF-TABLE is the lowest among the most recent baselines – 50% less than Binder and 75% less than Dater. We attribute the query efficiency of our method to the proposed dynamic operation execution through the tabular reasoning. The model is able to find an effective reasoning process that reaches the final output quicker and more reliably.

### 4.6 CASE STUDY

In Figure 4, we illustrate the tabular reasoning process by CHAIN-OF-TABLE. The question is based on a complex table and requires multiple reasoning steps to 1) identify the relevant columns, 2) conduct aggregation, and 3) reorder the aggregated intermediate information. Our proposed CHAIN-OF-TABLE involves dynamically planning an operation chain and accurately storing intermediate results in the transformed tables. These intermediate tables serve as tabular thought process that can guide the LLM to land to the correct answer more reliably.

## 5 CONCLUSION

Our proposed CHAIN-OF-TABLE enhances the reasoning capability of LLMs by leveraging the tabular structure to express intermediate thoughts for table-based reasoning. It instructs LLMs to dynamically plan an operation chain according to the input table and its associated question. This evolving table design sheds new light on the understanding of prompting LLMs for table understanding.

## 6 REPRODUCIBILITY STATEMENT

We include the prompt examples of DynamicPlan($T$,$Q$,chain) in Appendix E.1, the demo examples of GenerateArgs($T$,$Q$,f) in Appendix E.2, the prompt examples of Query($T$,$Q$) in Appendix E.3. We run the generic reasoning methods (End-to-End QA, FewShot QA, Chain-of-Thought) using the prompts reported in Appendix F. We run Text-to-SQL and Binder using the official open-sourced code and prompts in https://github.com/HKUNLP/Binder. We run Dater using the official open-sourced code and prompts in https://github.com/AlibabaResearch/DAMO-ConvAI. We revise the code to use publicly available GPT 3.5 and PaLM 2 (Section 4) as the LLM backbone instead of the OpenAI Codex due to its inaccessibility.

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

APPENDIX

# A  ATOMIC OPERATIONS IN CHAIN-OF-TABLE

## A.1  INTRODUCTION

In this study, we adopt a set of five table operations, which are commonly-used in SQL and DataFrame development, as an example. We note that our framework can trivially accommodate additional operations, which we leave for future work.

- `f_add_column()` adds a new column to the table to store intermediate reasoning or computational results.
- `f_select_row()` selects a subset of rows that are relevant to the question. Tables may contain irrelevant information for the given question (Ye et al., 2023). This operation helps locate the necessary context.
- `f_select_column()` selects a subset of columns. A column usually corresponds to an attribute in the table. This operation allows the model to locate the necessary attributes to answer the question.
- `f_group_by()` groups the rows by the contents of a specific column and provides the count of each enumeration value in that column. Many table-based questions or statements involve counting, but LLMs are not proficient at this task (Imani et al., 2023).
- `f_sort_by()` sorts the rows based on the contents of a specific column. When dealing with questions or statements involving comparison or extremes, LLMs can utilize this operation to rearrange the rows. The relationship can be readily inferred from the order of the sorted rows.

## A.2  ABLATION STUDY

To demonstrate the effectiveness of our proposed atomic operations, we perform an ablation study by creating five leave-one-out variants of our method, each of which removes one of the pre-defined operations from the pre-defined operation pool. For example, *w/o* `f_add_column()` means `f_add_column()` is removed from the operation pool. As a result, the LLM is only able to plan from the remaining four operations (`f_select_column`, `f_select_row`, `f_group_by`, and `f_sort_by`) to construct operation chains. We report the results of the ablation study in Table 5.

Table 5: Ablation study of the atomic operations used in CHAIN-OF-TABLE with PaLM 2 on WikiTQ and TabFact datasets. We observe that row selection and group-by operations have the biggest impact on the final table understanding performance.

| Prompting | TabFact | WikiTQ |
|---|---|---|
| | Accuracy | Accuracy |
| CHAIN-OF-TABLE | **86.61** | **67.31** |
| *w/o* `f_add_column()` | 85.23 (-1.38) | 65.88 (-1.43) |
| *w/o* `f_select_column()` | 82.61 (-4.00) | 65.68 (-1.63) |
| *w/o* `f_select_row()` | 82.21 (-4.40) | 65.06 (-2.25) |
| *w/o* `f_group_by()` | 84.78 (-1.83) | 61.88 (-5.43) |
| *w/o* `f_sort_by()` | 86.21 (-0.40) | 65.85 (-1.46) |

As shown in Table 5, all five operations contribute to the final state-of-the-art performance of CHAIN-OF-TABLE, as removing any operation results in a decrease in performance. In particular, we observe that `f_select_row()` and `f_select_column()` contribute the most on TabFact, while `f_group_by()` contributes the most on WikiTQ. This suggests that different tasks require different operations to help the LLM determine the correct answer. Therefore, leveraging the LLM to design custom operation chains through dynamic planning naturally fits different tasks, resulting in superior performance of our method.

## B EXPERIMENTS OF CHAIN-OF-TABLE ON FETAQA

Table 6 shows that CHAIN-OF-TABLE also improves the performance of free-form question answering on FeTaQA across all metrics, whereas Dater (Ye et al., 2023) fails to improve the ROUGE scores compared with End-to-End QA. We also observe the marginal improvement of CHAIN-OF-TABLE compared with the baseline methods. We attribute this to the nature of the n-gram text similarity metrics of ROUGE-1/2/L (Lin, 2004). As discussed in Maynez et al. (2023); Dhingra et al. (2019), these metrics are known to be insensitive to capturing improvements when using in-context learning since the model is unable to learn the expected style of the long form text just from an instruction or a few examples. We sample several cases from FeTaQA as shown in Figure 5 where the ROUGE metrics assign low scores; however, upon review, we observe that the generated answers were correct.

Table 6: Table understanding results on the FeTaQA benchmark using PaLM 2 with the best results in bold and improvements over Dater (Ye et al., 2023) reported. (underline denotes the second-best performance; **bold** denotes the best performance; the improvement is measured against the second-best performing method.)

| Prompting | FeTaQA | | | |
| --- | --- | --- | --- | --- |
| | BLEU | ROUGE-1 | ROUGE-2 | ROUGE-L |
| End-to-End QA | 28.37 | 0.63 | 0.41 | 0.53 |
| Dater (Ye et al., 2023) | 29.47 | 0.63 | 0.41 | 0.53 |
| CHAIN-OF-TABLE (ours) | **32.61** (+3.14) | **0.66** (+0.03) | **0.44** (+0.03) | **0.56** (+0.03) |

---

**Example from FeTaQA**

**Question**: Who were the last two finishers in the 2000 Summer Olympics Mens 100 metre freestyle?
**Answer**: Russia's Denis Pimankov (49.36) and Australia's Chris Fydler (49.44) rounded out the finale.
**Prediction**: The last two finishers in the 2000 Summer Olympics Mens 100 metre freestyle were Chris Fydler and Denis Pimankov.
**Results**: ROUGE-1=0.33; ROUGE-2=0.12; ROUGE-L=0.11

**Explanation**: The generated response correctly answers the question but the sentence styles are different. From the metrics, we can see the ROUGE scores are below the average.

---

Figure 5: Result example of CHAIN-OF-TABLE on FeTaQA using the ROUGE scores as metrics, where the ROUGE metrics assign very low scores but the generated answers were correct.

## C  INFERENCE PARAMETERS AND NUMBER OF DEMO SAMPLES OF CHAIN-OF-TABLE

We report the parameters and demo sample numbers we used in CHAIN-OF-TABLE in Table 7, 8 and 9. Overall, we annotate 29 samples and use them across different datasets. There are a large overlapping between the usage on different functions. For example, we use the same demo sample to introduce how to use `f_add_column` in the function `DynamicPlan` across different datasets. We guarantee that all demo samples are from the training set so they are unseen during testing. We argue that this further demonstrates our framework does not rely on a specific set of demos and can be well generalized to new datasets with the same prompts.

Table 7: LLM parameters and number of demo samples in CHAIN-OF-TABLE on WikiTQ

| Function | WikiTQ | | | | |
| --- | --- | --- | --- | --- | --- |
| | temperature | top_p | decode_steps | n_samples | n_demos |
| DynamicPlan() | 0.0 | 1.0 | 200 | - | 4 |
| f_add_column() | 0.0 | 1.0 | 200 | - | 6 |
| f_select_row() | 1.0 | 1.0 | 200 | 8 | 3 |
| f_select_column() | 1.0 | 1.0 | 200 | 8 | 8 |
| f_group_by() | 0.0 | 1.0 | 200 | - | 2 |
| f_sort_by() | 0.0 | 1.0 | 200 | - | 2 |
| query() | 0.0 | 1.0 | 200 | - | 1 |

Table 8: LLM parameters and number of demo samples in CHAIN-OF-TABLE on TabFact

| Function | TabFact | | | | |
| --- | --- | --- | --- | --- | --- |
| | temperature | top_p | decode_steps | n_samples | n_demos |
| DynamicPlan() | 0.0 | 1.0 | 200 | - | 4 |
| f_add_column() | 0.0 | 1.0 | 200 | - | 7 |
| f_select_row() | 0.5 | 1.0 | 200 | 8 | 4 |
| f_select_column() | 0.5 | 1.0 | 200 | 8 | 8 |
| f_group_by() | 0.0 | 1.0 | 200 | - | 2 |
| f_sort_by() | 0.0 | 1.0 | 200 | - | 2 |
| query() | 0.0 | 1.0 | 200 | - | 4 |

Table 9: LLM parameters and number of demo samples in CHAIN-OF-TABLE on FeTaQA

| Function | FeTaQA | | | | |
| --- | --- | --- | --- | --- | --- |
| | temperature | top_p | decode_steps | n_samples | n_demos |
| DynamicPlan() | 0.0 | 1.0 | 200 | - | 3 |
| f_add_column() | 0.0 | 1.0 | 200 | - | 6 |
| f_select_row() | 1.0 | 1.0 | 200 | 8 | 3 |
| f_select_column() | 1.0 | 1.0 | 200 | 8 | 8 |
| f_group_by() | 0.0 | 1.0 | 200 | - | 2 |
| f_sort_by() | 0.0 | 1.0 | 200 | - | 2 |
| query() | 0.0 | 1.0 | 200 | - | 8 |

## D    TABULAR FORMAT ENCODING COMPARISON

In alignment with prior studies Liu et al. (2023b; 2021); Jiang et al. (2022) and the baseline methods Cheng et al. (2022); Ye et al. (2023), we adopt PIPE encoding in CHAIN-OF-TABLE (as shown in Appendix E). This decouples the performance gains of the proposed tabular CoT with atomic operations from the influence of various table formatting choices.

To further understand the impact of different encoding methods on table understanding performance, we conduct additional experiments using 3 additional table representations: HTML, TSV, and Markdown. For these experiments, we use End-to-End QA on WikiTQ with PaLM 2 as a running example. The results are shown in Table 10. These findings show that different tabular format encoding methods lead to different outcomes. Notably, the PIPE format adopted in our study yields the highest performance among the four encoding methods tested.

Table 10: Tabular format encoding comparison on WikiTQ with PaLM 2

| Prompting | Tabular Format Encoding | | | |
| --- | --- | --- | --- | --- |
| | PIPE | HTML | TSV | Markdown |
| **End-to-End QA** | 60.6 | 56.1 | 58.1 | 58.0 |

## E    PROMPTS IN CHAIN-OF-TABLE

### E.1    DynamicPlan

We illustrate the prompting method used by DynamicPlan(T,Q,chain) in Figure 6 where $T$ is the latest intermediate table and $Q$ is its corresponding question; chain is the list of operations performed on the table.

With DynamicPlan, the LLM can generate the rest of the operation chain for the current sample (Figure 6(c)). We denote the generated operations as $f_{i+1}(\text{args}_{i+1}) \to ... \to [\text{E}]$ given that $f_i$ is the last operation of the input open-ended operation chain. Although a complete chain is generated, we only consider the first generated operation, $f_{i+1}$, and ignore the rest of the generation including the arguments and remaining operations. $f_{i+1}$ is generated based on the latest intermediate table from the previous operations, while the generation of subsequent operations are not based on the most up-to-date intermediate table so there could be mistakes in the generated contents. Therefore, we believe $f_{i+1}$ is the most reliable generation among all operations in the generated chain. See Figure 9 for more detailed prompts.

### E.2    GenerateArgs

We illustrate the demonstration examples and prompts used by GenerateArgs(T,Q,f) in Figure 7 where $T$ is the latest intermediate table and $Q$ is its corresponding question; f is the selected tabular operations. The detailed prompts for each operation and the regular expressions for extracting the generated arguments are as follows.

- f_add_column: See Figure 10.
- f_select_row: See Figure 12.
- f_select_column: See Figure 11.
- f_group_by: See Figure 13.
- f_sort_by: See Figure 14.

### E.3    Query

We illustrate the prompts used by Query(T,Q) in Figure 8 where $T$ is the resulting table from CHAIN-OF-TABLE and $Q$ is the question. See Figure 15 for more detailed prompts.

## F    IMPLEMENTATION DETAILS OF BASELINE METHODS

We run Text-to-SQL and Binder using the official open-sourced code and prompts in `https://github.com/HKUNLP/Binder`. We run Dater using the official open-sourced code and prompts in `https://github.com/AlibabaResearch/DAMO-ConvAI`. We revise the code to use publicly available GPT 3.5 and PaLM 2 (Section 4) as the LLM backbone instead of the OpenAI Codex due to its inaccessibility. We report the detailed prompts used in other baseline methods as follows.

- **End-to-End QA**: See Figure 16.
- **Few-Shot QA**: See Figure 17.
- **Chain-of-Thought**: The demonstration samples of Chain-of-Thought for WikiTQ and TabFact are from Chen (2023) (`https://github.com/wenhuchen/TableCoT`). See Figure 18.

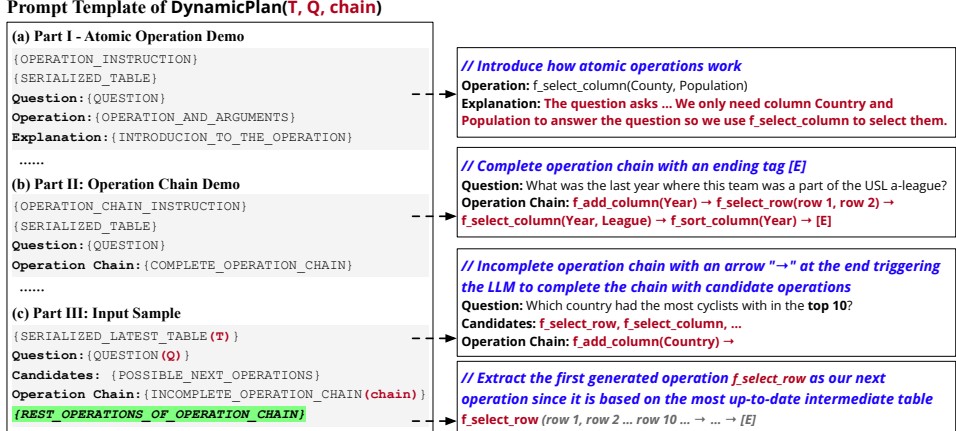

Figure 6:    Illustration of `DynamicPlan`($T$,$Q$,`chain`). **Left**: Overall prompt template and expected generation, including (a) demonstration of how atomic operations work, (b) demonstration of how to generate a complete operation chain to answer a given question, and (c) prompt for actual input table and its question, and its expected generation from the LLM (highlighted in green). **Right**: Examples and brief explanations of each part in the prompt and generation.

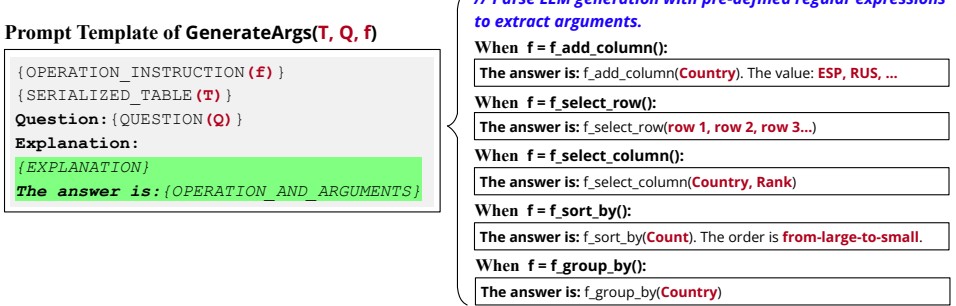

Figure 7:    Illustration of `GenerateArgs`($T$,$Q$,`f`). After a specific operation `f` is sampled by the LLM as the next operation, we ask the LLM to generate the required arguments by calling `GenerateArgs`. Then we parse the generation results of the LLM according to the pre-defined templates to extract the arguments.

**Prompt Template of Query(T, Q)**

*// Directly generate the answer with the resulting table
from Chain-of-Table*

```
{GENERAL_INSTRUCTION}
{SERIALIZED_RESULTING_TABLE(T)}
Question:{QUESTION(Q)}
Answer:{answer}
```

Figure 8: Illustration of Query($T, Q$). The resulting table from the operation chain serves as a proxy for the intermediate thoughts of reasoning, allowing us to directly generate the answer without providing the reasoning chain in textual format.

```
===================================== Prompt =====================================

If the table only needs a few rows to answer the question, we use f_select_row() to select
these rows for it. For example,
/*
col : Home team | Home Team Score | Away Team | Away Team Score | Venue | Crowd
row 1 : st kilda | 13.12 (90) | melbourne | 13.11 (89) | moorabbin oval | 18836
row 2 : south melbourne | 9.12 (66) | footscray | 11.13 (79) | lake oval | 9154
row 3 : richmond | 20.17 (137) | fitzroy | 13.22 (100) | mcg | 27651
*/
Question : Whose home team score is higher, richmond or st kilda?
Function: f_select_row(row 1, row 3)
Explanation: The question asks about the home team score of richmond and st kilda. We need
to know the the information of richmond and st kilda in row 1 and row 3. We select row 1
and row 3.

If the table only needs a few columns to answer the question, we use
f_select_column() to select these columns for it. For example,
......

If the question asks about items with the same value and the number of these items, we use
f_group_by() to group the items. For example,
......

If the question asks about the order of items in a column, we use f_sort_by() to sort
the items. For example,
......

Here are examples of using the operations to answer the question.
/*
col : Date | Division | League | Regular Season | Playoffs | Open Cup
row 1 : 2001/01/02 | 2 | USL A-League | 4th, Western | Quarterfinals | Did not qualify
row 2 : 2002/08/06 | 2 | USL A-League | 2nd, Pacific | 1st Round | Did not qualify
row 5 : 2005/03/24 | 2 | USL First Division | 5th | Quarterfinals | 4th Round
*/
Question: what was the last year where this team was a part of the usl a-league?
Function Chain: f_add_column(Year) -> f_select_row(row 1, row 2) ->
f_select_column(Year, League) -> f_sort_by(Year) -> <END>
......

/*
col : Rank | Cyclist | Team | Time | UCI ProTour; Points | Country
row 1 : 1 | Alejandro Valverde (ESP) | Caisse d'Epargne | 5h 29' 10" | 40 | ESP
row 2 : 2 | Alexandr Kolobnev (RUS) | Team CSC Saxo Bank | s.t. | 30 | RUS
row 3 : 3 | Davide Rebellin (ITA) | Gerolsteiner | s.t. | 25 | ITA
row 4 : 4 | Paolo Bettini (ITA) | Quick Step | s.t. | 20 | ITA
row 5 : 5 | Franco Pellizotti (ITA) | Liquigas | s.t. | 15 | ITA
row 6 : 6 | Denis Menchov (RUS) | Rabobank | s.t. | 11 | RUS
row 7 : 7 | Samuel Sánchez (ESP) | Euskaltel-Euskadi | s.t. | 7 | ESP
row 8 : 8 | Stéphane Goubert (FRA) | Ag2r-La Mondiale | + 2" | 5 | FRA
row 9 : 9 | Haimar Zubeldia (ESP) | Euskaltel-Euskadi | + 2" | 3 | ESP
row 10 : 10 | David Moncoutié (FRA) | Cofidis | + 2" | 1 | FRA
*/
Question: which country had the most cyclists finish within the top 10?
The next operation must be one of f_select_row() or f_select_column() or f_group_by()
or f_sort_by().
Function Chain: f_add_column(Country) ->

=================================== Completion ===================================

f_select_row(row 1, row 10) -> f_select_column(Country) -> f_group_by(Country) -> <END>
```

Figure 9: DynamicPlan(T,Q,chain) Prompt used for WikiTQ

```
To answer the question, we can first use f_add_column() to add more columns to the table.

The added columns should have these data types:
1. Numerical: the numerical strings that can be used in sort, sum
2. Datetype: the strings that describe a date, such as year, month, day
3. String: other strings

/*
col : Week | When | Kickoff | Opponent | Results; Final score | Results; Team record
row 1 : 1 | Saturday, April 13 | 7:00 p.m. | at Rhein Fire | W 27-21 | 1-0
row 2 : 2 | Saturday, April 20 | 7:00 p.m. | London Monarchs | W 37-3 | 2-0
row 3 : 3 | Sunday, April 28 | 6:00 p.m. | at Barcelona Dragons | W 33-29 | 3-0
*/
Question: what is the date of the competition with highest attendance?
The existing columns are: "Week", "When", "Kickoff", "Opponent", "Results; Final score",
"Results; Team record", "Game site", "Attendance".
Explanation: the question asks about the date of the competition with highest score. Each
row is about one competition. We extract the value from column "Attendance" and create a
different column "Attendance number" for each row. The datatype is Numerical.
Therefore, the answer is: f_add_column(Attendance number). The value: 32092 | 34186 | 17503

/*
col : Rank | Lane | Player | Time
row 1 :  | 5 | Olga Tereshkova (KAZ) | 51.86
row 2 :  | 6 | Manjeet Kaur (IND) | 52.17
row 3 :  | 3 | Asami Tanno (JPN) | 53.04
*/
Question: tell me the number of athletes from japan.
The existing columns are: Rank, Lane, Player, Time.
Explanation: the question asks about the number of athletes from japan. Each row is about
one athlete. We need to know the country of each athlete. We extract the value from column
"Player" and create a different column "Country of athletes" for each row. The datatype
is String.
Therefore, the answer is: f_add_column(Country of athletes). The value: KAZ | IND | JPN
```

Figure 10: Demos used for `GenerateArgs(T,Q,f_add_column)`. We use the regular expression: `f_add_column((.*)).The value:(.*)` to extract the arguments from the generated text.

```
Use f_select_column() to filter out useless columns in the table according to information
in the statement and the table.

/*
{
  "table_caption": "south wales derby",
  "columns": ["competition", "total matches", "cardiff win", "draw", "swansea win"],
  "table_column_priority": [
    ["competition", "league", "fa cup", "league cup"],
    ["total matches", "55", "2", "5"],
    ["cardiff win", "19", "0", "2"],
    ["draw", "16", "27", "0"],
    ["swansea win", "20", "2", "3"]
  ]
}
*/
statement : there are no cardiff wins that have a draw greater than 27.
similar words link to columns :
no cardiff wins -> cardiff win
a draw -> draw
column value link to columns :
27 -> draw
semantic sentence link to columns :
None
The answer is : f_select_column([cardiff win, draw])
```

Figure 11: Demos used for `GenerateArgs(T,Q,f_select_column)`. We use the regular expression: `f_select_column([(.*)])` to extract the arguments from the generated text.

```
Using f_select_row() to select relevant rows in the given table that support or oppose the
statement.
Please use f_select_row([*]) to select all rows in the table.

/*
table caption : 1972 vfl season.
col : home team | home team score | away team | away team score | venue | crowd
row 1 : st kilda | 13.12 (90) | melbourne | 13.12 (89) | moorabbin oval | 18836
row 2 : south melbourne | 9.12 (66) | footscray | 11.13 (79) | lake oval | 9154
row 3 : richmond | 20.17 (137) | fitzroy | 13.22 (100) | mcg | 27651
row 4 : geelong | 17.10 (112) | collingwood | 17.9 (111) | kardinia park | 23108
row 5 : north melbourne | 8.12 (60) | carlton | 23.11 (149) | arden street oval | 11271
row 6 : hawthorn | 15.16 (106) | essendon | 12.15 (87) | vfl park | 36749
*/
statement : what is the away team with the highest score?
explain : the statement want to ask the away team of highest away team score. the highest
away team score is 23.11 (149). it is on the row 5.so we need row 5.
The answer is : f_select_row([row 5])
```

Figure 12: Demos used for GenerateArgs(T,Q,f_select_row). We use the regular expression: f_select_row([(.*)]) to extract the arguments from the generated text.

```
To answer the question, we can first use f_group_by() to group the values in a column.

/*
col : Rank | Lane | Athlete | Time | Country
row 1 : 1 | 6 | Manjeet Kaur (IND) | 52.17 | IND
row 2 : 2 | 5 | Olga Tereshkova (KAZ) | 51.86 | KAZ
row 3 : 3 | 4 | Pinki Pramanik (IND) | 53.06 | IND
row 4 : 4 | 1 | Tang Xiaoyin (CHN) | 53.66 | CHN
row 5 : 5 | 8 | Marina Maslyonko (KAZ) | 53.99 | KAZ
*/
Question: tell me the number of athletes from japan.
The existing columns are: Rank, Lane, Athlete, Time, Country.
Explanation: The question asks about the number of athletes from India. Each row is about
an athlete. We can group column "Country" to group the athletes from the same country.
Therefore, the answer is: f_group_by(Country).
```

Figure 13: Demos used for GenerateArgs(T,Q,f_group_by). We use the regular expression: f_group_by((.*)) to extract the arguments from the generated text.

```
To answer the question, we can first use f_sort_by() to sort the values in a column to get
the
order of the items. The order can be "large to small" or "small to large".

The column to sort should have these data types:
1. Numerical: the numerical strings that can be used in sort
2. DateType: the strings that describe a date, such as year, month, day
3. String: other strings

/*
col : Position | Club | Played | Points | Wins | Draws | Losses | Goals for | Goals against
row 1 : 1 | Malaga CF | 42 | 79 | 22 | 13 | 7 | 72 | 47
row 10 : 10 | CP Merida | 42 | 59 | 15 | 14 | 13 | 48 | 41
row 3 : 3 | CD Numancia | 42 | 73 | 21 | 10 | 11 | 68 | 40
*/
Question: what club placed in the last position?
The existing columns are: Position, Club, Played, Points, Wins, Draws, Losses, Goals for,
Goals against
Explanation: the question asks about the club in the last position. Each row is about a
club. We need to know the order of position from last to front. There is a column for
position and the column name is Position. The datatype is Numerical.
Therefore, the answer is: f_sort_by(Position), the order is "large to small".
```

Figure 14: Demos used for GenerateArgs(T,Q,f_sort_by). We use the regular expression: f_sort_by((.*)),the order is "(.*)". to extract the arguments from the generated text.

```
===================================== Prompt =====================================

Here is the table to answer this question. Please understand the table and answer the
question:

/*
col : Rank | City | Passengers Number | Ranking | Airline
row 1 : 1 | United States, Los Angeles | 14749 | 2 | Alaska Airlines
row 2 : 2 | United States, Houston | 5465 | 8 | United Express
row 3 : 3 | Canada, Calgary | 3761 | 5 | Air Transat, WestJet
row 4 : 4 | Canada, Saskatoon | 2282 | 4 |
row 5 : 5 | Canada, Vancouver | 2103 | 2 | Air Transat
row 6 : 6 | United States, Phoenix | 1829 | 1 | US Airways
row 7 : 7 | Canada, Toronto | 1202 | 1 | Air Transat, CanJet
row 8 : 8 | Canada, Edmonton | 110 | 2 |
row 9 : 9 | United States, Oakland | 107 | 5 |
*/
Question: how many more passengers flew to los angeles than to saskatoon from manzanillo
airport in 2013?
The anwser is: 12467

Here is the table to answer this question. Please understand the table and answer the
question:

/*
col : Rank | Country
row 1 : 1 | ESP
row 2 : 2 | RUS
row 3 : 3 | ITA
row 4 : 4 | ITA
row 5 : 5 | ITA
row 6 : 6 | RUS
row 7 : 7 | ESP
row 8 : 8 | FRA
row 9 : 9 | ESP
row 10 : 10 | FRA
*/
Group the rows according to column "Country":
/*
Group ID | Country | Count
1 | ITA | 3
2 | ESP | 3
3 | RUS | 2
4 | FRA | 2
*/
Question: which country had the most cyclists in top 10?
The answer is:

=================================== Completion ===================================
Italy.
```

Figure 15: Prompt Example used for Query(T,Q)

```
======================================== Prompt ========================================

Here is the table to answer this question. Answer the question.
/*
col : Name | League | FA Cup | League Cup | JP Trophy | Total
row 1 : Scot Bennett | 5 | 0 | 0 | 0 | 5
row 2 : Danny Coles | 3 | 0 | 0 | 0 | 3
row 3 : Liam Sercombe | 1 | 0 | 0 | 0 | 1
row 4 : Alan Gow | 4 | 0 | 0 | 0 | 4
row 5 : John O'Flynn | 11 | 0 | 1 | 0 | 12
row 6 : Guillem Bauza | 2 | 0 | 0 | 0 | 2
row 7 : Jimmy Keohane | 3 | 0 | 0 | 0 | 3
row 8 : Pat Baldwin | 1 | 0 | 0 | 0 | 1
row 9 : Jamie Cureton | 20 | 0 | 0 | 0 | 20
row 10 : Arron Davies | 3 | 0 | 0 | 0 | 3
row 11 : Jake Gosling | 1 | 0 | 0 | 0 | 1
row 12 : OWN GOALS | 0 | 0 | 0 | 0 | 0
row 13 : Total | 0 | 0 | 0 | 0 | 0
*/
Question: does pat or john have the highest total?
The answer is:

====================================== Completion ======================================

John.
```

Figure 16: Prompt of End-to-end QA used for WikiTQ.

```
==================================== Prompt ====================================

Here is the table to answer this question. Answer the question.
/*
col : Rank | Cyclist | Team | Time | UCI ProTour; Points
row 1 : 1 | Alejandro Valverde (ESP) | Caisse d'Epargne | 5h 29' 10" | 40
row 2 : 2 | Alexandr Kolobnev (RUS) | Team CSC Saxo Bank | s.t. | 30
row 3 : 3 | Davide Rebellin (ITA) | Gerolsteiner | s.t. | 25
row 4 : 4 | Paolo Bettini (ITA) | Quick Step | s.t. | 20
row 5 : 5 | Franco Pellizotti (ITA) | Liquigas | s.t. | 15
row 6 : 6 | Denis Menchov (RUS) | Rabobank | s.t. | 11
row 7 : 7 | Samuel Sánchez (ESP) | Euskaltel-Euskadi | s.t. | 7
row 8 : 8 | Stéphane Goubert (FRA) | Ag2r-La Mondiale | + 2" | 5
row 9 : 9 | Haimar Zubeldia (ESP) | Euskaltel-Euskadi | + 2" | 3
row 10 : 10 | David Moncoutié (FRA) | Cofidis | + 2" | 1
*/
Question: which country had the most cyclists finish within the top 10?
The answer is: Italy.

Here is the table to answer this question. Please provide your explanation first, then
answer the question in a short phrase starting by 'therefore, the answer is:'
/*
col : Rank | Cyclist | Team | Time | UCI ProTour; Points
row 1 : 1 | Alejandro Valverde (ESP) | Caisse d'Epargne | 5h 29' 10" | 40
row 2 : 2 | Alexandr Kolobnev (RUS) | Team CSC Saxo Bank | s.t. | 30
row 3 : 3 | Davide Rebellin (ITA) | Gerolsteiner | s.t. | 25
row 4 : 4 | Paolo Bettini (ITA) | Quick Step | s.t. | 20
row 5 : 5 | Franco Pellizotti (ITA) | Liquigas | s.t. | 15
row 6 : 6 | Denis Menchov (RUS) | Rabobank | s.t. | 11
row 7 : 7 | Samuel Sánchez (ESP) | Euskaltel-Euskadi | s.t. | 7
row 8 : 8 | Stéphane Goubert (FRA) | Ag2r-La Mondiale | + 2" | 5
row 9 : 9 | Haimar Zubeldia (ESP) | Euskaltel-Euskadi | + 2" | 3
row 10 : 10 | David Moncoutié (FRA) | Cofidis | + 2" | 1
*/
Question: how many players got less than 10 points?
The answer is: 4.

Here is the table to answer this question. Answer the question.
/*
col : Name | League | FA Cup | League Cup | JP Trophy | Total
row 1 : Scot Bennett | 5 | 0 | 0 | 0 | 5
row 2 : Danny Coles | 3 | 0 | 0 | 0 | 3
row 3 : Liam Sercombe | 1 | 0 | 0 | 0 | 1
row 4 : Alan Gow | 4 | 0 | 0 | 0 | 4
row 5 : John O'Flynn | 11 | 0 | 1 | 0 | 12
row 6 : Guillem Bauza | 2 | 0 | 0 | 0 | 2
row 7 : Jimmy Keohane | 3 | 0 | 0 | 0 | 3
row 8 : Pat Baldwin | 1 | 0 | 0 | 0 | 1
row 9 : Jamie Cureton | 20 | 0 | 0 | 0 | 20
row 10 : Arron Davies | 3 | 0 | 0 | 0 | 3
row 11 : Jake Gosling | 1 | 0 | 0 | 0 | 1
row 12 : OWN GOALS | 0 | 0 | 0 | 0 | 0
row 13 : Total | 0 | 0 | 0 | 0 | 0
*/
Question: does pat or john have the highest total?
The answer is:

================================== Completion ==================================

John.
```

Figure 17: Prompt of Few-shot QA used for WikiTQ

```
======================================== Prompt ========================================
Here is the table to answer this question. Please provide your explanation first, then
answer the question in a short phrase starting by 'therefore, the answer is:'
/*
col : Rank | Cyclist | Team | Time | UCI ProTour; Points
row 1 : 1 | Alejandro Valverde (ESP) | Caisse d'Epargne | 5h 29' 10" | 40
row 2 : 2 | Alexandr Kolobnev (RUS) | Team CSC Saxo Bank | s.t. | 30
row 3 : 3 | Davide Rebellin (ITA) | Gerolsteiner | s.t. | 25
row 4 : 4 | Paolo Bettini (ITA) | Quick Step | s.t. | 20
row 5 : 5 | Franco Pellizotti (ITA) | Liquigas | s.t. | 15
row 6 : 6 | Denis Menchov (RUS) | Rabobank | s.t. | 11
row 7 : 7 | Samuel Sánchez (ESP) | Euskaltel-Euskadi | s.t. | 7
row 8 : 8 | Stéphane Goubert (FRA) | Ag2r-La Mondiale | + 2" | 5
row 9 : 9 | Haimar Zubeldia (ESP) | Euskaltel-Euskadi | + 2" | 3
row 10 : 10 | David Moncoutié (FRA) | Cofidis | + 2" | 1
*/
Question: which country had the most cyclists finish within the top 10?
Explanation: ITA occurs three times in the table, more than any others. Therefore, the
answer is: Italy.

Here is the table to answer this question. Please provide your explanation first, then
answer the question in a short phrase starting by 'therefore, the answer is:'
/*
col : Rank | Cyclist | Team | Time | UCI ProTour; Points
row 1 : 1 | Alejandro Valverde (ESP) | Caisse d'Epargne | 5h 29' 10" | 40
row 2 : 2 | Alexandr Kolobnev (RUS) | Team CSC Saxo Bank | s.t. | 30
row 3 : 3 | Davide Rebellin (ITA) | Gerolsteiner | s.t. | 25
row 4 : 4 | Paolo Bettini (ITA) | Quick Step | s.t. | 20
row 5 : 5 | Franco Pellizotti (ITA) | Liquigas | s.t. | 15
row 6 : 6 | Denis Menchov (RUS) | Rabobank | s.t. | 11
row 7 : 7 | Samuel Sánchez (ESP) | Euskaltel-Euskadi | s.t. | 7
row 8 : 8 | Stéphane Goubert (FRA) | Ag2r-La Mondiale | + 2" | 5
row 9 : 9 | Haimar Zubeldia (ESP) | Euskaltel-Euskadi | + 2" | 3
row 10 : 10 | David Moncoutié (FRA) | Cofidis | + 2" | 1
*/
Question: how many players got less than 10 points?
Explanation: Samuel Sánchez,  Stéphane Goubert, Haimar Zubeldia and David Moncoutié
received less than 10 points.  Therefore, the answer is: 4.

Here is the table to answer this question. Please provide your explanation first, then
answer the question in a short phrase starting by 'therefore, the answer is:'
/*
col : Name | League | FA Cup | League Cup | JP Trophy | Total
row 1 : Scot Bennett | 5 | 0 | 0 | 0 | 5
row 2 : Danny Coles | 3 | 0 | 0 | 0 | 3
row 3 : Liam Sercombe | 1 | 0 | 0 | 0 | 1
row 4 : Alan Gow | 4 | 0 | 0 | 0 | 4
row 5 : John O'Flynn | 11 | 0 | 1 | 0 | 12
row 6 : Guillem Bauza | 2 | 0 | 0 | 0 | 2
row 7 : Jimmy Keohane | 3 | 0 | 0 | 0 | 3
row 8 : Pat Baldwin | 1 | 0 | 0 | 0 | 1
row 9 : Jamie Cureton | 20 | 0 | 0 | 0 | 20
row 10 : Arron Davies | 3 | 0 | 0 | 0 | 3
row 11 : Jake Gosling | 1 | 0 | 0 | 0 | 1
row 12 : OWN GOALS | 0 | 0 | 0 | 0 | 0
row 13 : Total | 0 | 0 | 0 | 0 | 0
*/
Question: does pat or john have the highest total?
Explanation:
====================================== Completion ======================================
John O'Flynn has the highest total of 12 goals. Pat Baldwin has the lowest total of 1 goal.
Therefore, the answer is: John.
```

Figure 18: Prompt of Chain-of-Thought used for WikiTQ

