# OpenReview forum: "Chain-of-Table: Evolving Tables in the Reasoning Chain for Table Understanding"
_ICLR.cc/2024/Conference — ICLR 2024 poster_

### Official Review · Reviewer_xr4a · 2023-10-30

**Soundness:** 3 good
**Presentation:** 3 good
**Contribution:** 2 fair
**Rating:** 5
**Confidence:** 3

**Summary:**

This paper proposes chain-of-tables prompting for solving table-based reasoning tasks. Specifically, instead of generating reasoning chains in natural language or code, the paper generates reasoning chains as a list of atomic operations on the table, and use the intermediate updated table to represent the reasoning process. LLMs are used to generate the operation chains, and each operation is done by executing it in some programming languages. Experiments show that the proposed method outperforms existing baselines on three benchmarks.

**Strengths:**

1. The proposed method is new in that it adapts the previous chain-of-thought prompting to the table-related tasks, and it improves the understanding of intermediate reasoning results.
2. Experiments on three benchmarks show the advantages of proposed method over a broad range of baselines including end-to-end methods and other program-aided methods.

**Weaknesses:**

1. The proposed method requires significantly more number of queries to LLMs compared to baselines such as chain-of-thought and Binder, since the method needs to query LLMs twice (one for operation generation and one for arguments generation) for each operation. However, no comparison is provided for performance under the same number of queries. For example, what is the performance of baselines when they adopt the self-consistency idea (which is shown to be beneficial in the Binder paper)? Does the proposed method still have advantage when using the same number of queries?
2. The pre-defined five atomic operations seem to significantly limit the available operations. For example, aggregations such as SUM, MAX, MIN, etc. can be easily done in SQL. How are these aggregations done in the proposed method? No explanation is provided for why these five operations are used in the paper and why they cover the complete set of operations on table.
3. The presentation of argument generation process is not clear. Based on the prompts in the Appendix, it seems add_column operation directly uses LLMs to generate the new column, whereas other four operations only prompt LLMs to generate arguments that will be fed to the programming language.

**Questions:**

1. The added API calls in Binder also add additional columns to the table, how is the proposed method different from the operation used in Binder?

---

> ### Author Response · Authors · 2023-11-22
> **Response to reviewer xr4a (Part 1)**
>
> We thank the reviewer for sharing the concerns and pointing out several issues in our current submission. The reviewer has questions about the number of queries, reason for adopting the 5 operations, argument generation process, and add_column in Binder. We address these points carefully in the individual sections below.
>
>
>
>
>
>
> > The proposed method requires significantly more number of queries to LLMs compared to baselines
>
> **[Response]** Thanks for the question. We note that the number of queries required by our method is actually less than the competing methods. This is because Chain-of-Table employs a tabular reasoning process to decompose complex queries and tables, instead of relying on self-consistency sampling for boosting performance. This resulted in a reduced query count for our method. See detailed statistics below:
>
> *[Experiment on TabFact with PaLM 2]*
>
> | Method                | Total Sampling Steps | Details                                                    | Accuracy |
> | --------------------- | -------------------- | ---------------------------------------------------------- | -------- |
> | Binder                | 50                   | Generate Neural-SQL: 50                                    | 76.98    |
> | Dater                 | 80                   | TableDecompose: 40; Cloze: 20; Generate SQL: 20; Query: 20 | 84.63    |
> | Chain-of-Table (ours) | **25**                   | DynamicPlan: 5 max; GenerateArgs: 19 max; Query: 1         | 86.61    |
>
> We observe that the number of queries needed by Chain-of-Table is the lowest among the most recent baselines - 50% less than Binder and 68.75% less than Dater. We attribute the query efficiency of our method to the proposed dynamic operation execution through the tabular reasoning. The model is able to find an effective iterative process [A1, A2, A3] that reaches the final output quicker and more reliably.
>
> - [A1] Khot, Tushar, et al. "Decomposed Prompting: A Modular Approach for Solving Complex Tasks." The Eleventh International Conference on Learning Representations. 2022.
> - [A2] Zhou, Denny, et al. "Least-to-Most Prompting Enables Complex Reasoning in Large Language Models." The Eleventh International Conference on Learning Representations. 2022.
> - [A3] Yao, Shunyu, et al. "ReAct: Synergizing Reasoning and Acting in Language Models." The Eleventh International Conference on Learning Representations. 2022.

---

> ### Author Response · Authors · 2023-11-22
> **Response to reviewer xr4a (Part 2)**
>
> > No explanation is provided for why these five operations are used in the paper and why they cover the complete set of operations on table.
>
> **[Response]** Thanks for the questions regarding the adaptability of the operation pool and the rationale behind the five operations used in this work.
>
> We observe that the previous works either can not adopt new operations or is limited to a certain type of operations. Specifically, Dater [A4] employs a dedicated and fixed procedure for decomposing tables and questions, which lacks compatibility with new operations. Binder [A5], while potentially compatible with new operations, is limited to those compatible with code interpreters such as SQL or Python.
>
> In contrast, the proposed Chain-of-Table represents the first initiative to overcome these limitations. It enables the adoption of an arbitrary number of new operations through flexible in-context demonstrations, without the need for operations to adhere to strict program executor formats.
>
> Following this motivation, we validate the flexibility and efficacy of the proposed Chain-of-Table by simply selecting the most frequently used table operations as reported in (https://bytescout.com/blog/20-important-sql-queries.html). The operation pool in our method is adaptable and can be customized if prior knowledge of the queries and/or input tables is available.
>
> - [A4] Ye, Yunhu, et al. "Large Language Models are Versatile Decomposers: Decomposing Evidence and Questions for Table-based Reasoning." Proceedings of the 46th International ACM SIGIR Conference on Research and Development in Information Retrieval. 2023.
> - [A5] Cheng, Zhoujun, et al. "Binding Language Models in Symbolic Languages." The Eleventh International Conference on Learning Representations. 2022.

---

> ### Author Response · Authors · 2023-11-22
> **Response to reviewer xr4a (Part 3)**
>
> > Add_column operation uses LLMs?
>
> **[Response]** That's correct. Our Chain-of-Table uses the proposed tabular reasoning to decide whether to use add_column given the query and the input table. The output of the add_column operation is generated by LLMs.

---

> ### Author Response · Authors · 2023-11-22
> **Response to reviewer xr4a (Part 4)**
>
> > The added API calls in Binder also add additional columns to the table, how is the proposed method different from the operation used in Binder?
>
> **[Response]** Thanks for the question and the opportunity to provide further clarification. There are two major differences between the two methods.
>
> First, the add_column operation in Binder requires SQL execution, inherently limits the computation to the capabilities of SQL itself. In contrast, our Chain-of-Table method employs LLMs to generate the contents for the *add_column* operation without further calling a SQL executor. This approach allows us to go beyond the SQL requirements and utilize the internal knowledge and generalization capabilities of LLM to generate the output without constraints.
>
> Second, Binder operates as a single-step pipeline, so the generated SQL queries do not have knowledge of the newly added column. On the other hand, Chain-of-Table benefits from an iterative process where the *add_column* operation produces intermediate tables, thereby eliciting the following reasoning steps. Therefore, Binder represents a special case of the Chain-of-Table framework, applicable when the pipeline comprises only a single step.

---

### Official Review · Reviewer_3YkD · 2023-11-01

**Soundness:** 3 good
**Presentation:** 3 good
**Contribution:** 3 good
**Rating:** 6
**Confidence:** 3

**Summary:**

The paper aims to effectively leverage tabular data in the reasoning chain. To achieve this, the paper proposes the chain-of-table that conducts step-by-step reasoning as step-by-step tabular operations to form a chain of tables. Empirical results on three benchmarks verify the effectiveness of the proposed method.

**Strengths:**

The investigated problem of leveraging and understanding structural data like tables is important and practical while existing LLMs can not solve it well.

The proposed method does not require training (or fine-tuning) of the existing LLMs.

The design of atomic operations is novel and also reasonable.

The overall reasoning procedure of chain-of-table is step-by-step, explainable, and effective.

The paper is solid from a technical perspective, and extensive experiments are conducted.

The presentation and drawn figures are generally clear and easy to understand.

Several case studies are also elaborated on in the Appendix.

**Weaknesses:**

The proposed method only achieves marginal improvements in some cases, e.g., TabFact and ROUGE-1/2/L datasets. I would suggest the paper discuss the potential reasons.

The observations in Figure 4 are quite interesting. It seems that a longer chain does not consistently bring more accurate results. What are the underlying reasons for this?

Dater (Ye et al., 2023) should be the most important baseline for comparison. I would suggest the paper make a further comparison with in-depth analysis from the perspective of methodology, e.g., a comparison between one-step and multi-step reasoning on tabular data.

Besides, how efficient is chain-of-tables when dealing with large-scale data? It seems that the running-time efficiency is known from the current draft.

**Questions:**

Please refer to the above weakness part.

---

> ### Author Response · Authors · 2023-11-21
> **Response to reviewer 3YkD (Part 1)**
>
> We would like to thank the reviewer 3YkD for the insightful comments and suggestions. Please see response below:
>
> > The proposed method only achieves marginal improvements in some cases, e.g., TabFact and ROUGE-1/2/L datasets. I would suggest the paper discuss the potential reasons.
>
> **[Response]** Thanks for pointing this out. We attribute this marginal improvement to the nature of the n-gram text similarity metrics of ROUGE-1/2/L. As discussed in [A1, A2], these metrics are known to be insensitive to capturing improvements when using in-context learning since the model is unable to learn the expected style of the long form text just from an instruction or a few examples. We sampled several cases from FeTaQA where the ROUGE metrics assigned very low scores; however, upon review, we observed that the generated answers were correct.
>
> **ID: 9580**
>
> - **Question:** Who were the last two finishers in the 2000 Summer Olympics Mens 100 metre freestyle?
> - **Answer:** Russia's *Denis Pimankov* (49.36) and Australia's *Chris Fydler* (49.44) rounded out the finale.
> - **Prediction:** The last two finishers in the 2000 Summer Olympics Mens 100 metre freestyle were *Chris Fydler* and *Denis Pimankov*.
> - **ROUGE-1:** 0.33
> - **ROUGE-2:** 0.12
> - **ROUGE-L:** 0.11
> - **Explanation:** The generated response correctly answers the question but the sentence styles are different. From the metrics, we can see the ROUGE scores are below the average (On average, Chain-of-Table achieves ROUGE-1=0.66; ROUGE-2=0.44; ROUGE-L=0.56).
>
> **ID: 9299**
>
> - **Question:** Were any NR times set in the Women's 800 metre freestyle final?
> - **Answer:** *Yana Klochkova of Ukraine and Flavia Rigamonti of Switzerland both set NR times* of 8:22.66 and 8:25.91 respectively.
> - **Prediction:** *Yes, two NR times* were set in the Women's 800 metre freestyle final.
> - **ROUGE-1:** 0.17
> - **ROUGE-2:** 0.06
> - **ROUGE-L:** 0.11
> - **Explanation:** The question is actually a "Yes/No question" and we believe our generated answer is sufficient to it. However, the groundtruth answer provides more details and thus has a distinct sentence style. The n-gram metrics cannot catch this and give a low ROUGH-score.
>
> Nevertheless, the proposed Chain-of-Table still shows noticeable improvement while the top competing method Dater does not show improvement. We will discuss this in the final version.
>
> - [A1] Maynez, Joshua, Priyanka Agrawal, and Sebastian Gehrmann. "Benchmarking Large Language Model Capabilities for Conditional Generation." Proceedings of the 61st Annual Meeting of the Association for Computational Linguistics (Volume 1: Long Papers). 2023.
> - [A2] Dhingra, Bhuwan, et al. "Handling Divergent Reference Texts when Evaluating Table-to-Text Generation." Proceedings of the 57th Annual Meeting of the Association for Computational Linguistics. 2019.

---

> ### Author Response · Authors · 2023-11-21
> **Response to reviewer 3YkD (Part 2)**
>
> > The observations in Figure 4 are quite interesting. It seems that a longer chain does not consistently bring more accurate results. What are the underlying reasons for this?
>
> **[Response]** We appreciate the reviewer's interest in our analysis presented in Figure 4. We also observe that longer chains do not necessarily equate to better performance. From our perspective, this trend is consistent with our expectations.
>
> The presence of longer chains suggests that the questions and their corresponding input table are more complex and challenging to solve, as Chain-of-Table dynamically plans the operation chain in response to the question's needs (as detailed in Section 4.4). Consequently, the length of these chains does not directly correlate with increased accuracy in the results.

---

> ### Author Response · Authors · 2023-11-21
> **Response to reviewer 3YkD (Part 3)**
>
> > Dater (Ye et al., 2023) should be the most important baseline for comparison. I would suggest the paper make a further comparison with in-depth analysis from the perspective of methodology, e.g., a comparison between one-step and multi-step reasoning on tabular data.
>
> **[Response]** Dater is the top competing baseline. Here we further compare Dater and the proposed Chain-of-Table.
>
> Both Dater and Chain-of-Table tackle table understanding by decomposing input tabular data. However, Dater is confined to row and column selection operations, and these operations are always executed irrespective of their necessity. Moreover, the primary purpose of these row and column selection operations in Dater is to serve as a pre-processing step for tables, reducing the cell count before they are fed to the LLMs.
>
> In contrast, our Chain-of-Table method provides an extendable framework capable of incorporating an arbitrary number of operations. We incorporate the 5 operations in the paper for simplicity and demonstration. The primary goal of these operations is to perform tabular reasoning. This is achieved by adaptively selecting and executing operations according to the difficulty and complexity of the query and the corresponding input table. We will include these details in our final version.

---

> ### Author Response · Authors · 2023-11-21
> **Response to reviewer 3YkD (Part 4)**
>
> > Besides, how efficient is chain-of-tables when dealing with large-scale data?
>
> **[Response]** We follow the suggestion to compute the total number of input and output tokens required for Chain-of-Table compared to the top competing method, Dater, as shown in the table below. The 2nd column indicates the total tokens required, while the 3rd and 4th columns provide a detailed breakdown of the calculation process. The numbers in brackets in the 3rd and 4th columns represent the number of samples taken for self-consistency. We sample 20 examples from WikiTQ and use PaLM 2 as the backbone. The numbers below are computed on this subset.
>
> | Method         | Total input tokens per sample / Total output tokens per sample | Est. details of input tokens per sample                                 | Est. details of output tokens per sample                                           |
> | -------------- | ------------------------------------------------ | ----------------------------------------------------------------------- | ---------------------------------------------------------------------------------- |
> | Chain-of-Table | 12130 / 656                                      | DynamicPlan: 1654 (1 ~ 5); GenerateArgs: 445 ~ 2598; Query: 229           | DynamicPlan: 47; GenerateArgs: 47 ~ 86; Query: 3                                     |
> | Dater          | 33916 / 4360                                     | TableDecompose: 3774; Cloze: 927; Generate SQL: 1424 (x 20); Query: 735 | TableDecompose: 148 (x 20); Cloze: 30 (x 20); Generate SQL: 36 (x 20); Query: 4 (x 20) |
>
> The results show that the Chain-of-Table method requires significantly fewer input and output tokens compared to Dater, demonstrating its efficiency in token usage during the reasoning process.

---

### Official Review · Reviewer_eSFE · 2023-11-02

**Soundness:** 3 good
**Presentation:** 3 good
**Contribution:** 2 fair
**Rating:** 5
**Confidence:** 4

**Summary:**

The paper introduces a novel framework called "chain-of-table" for reasoning on tabular data. This framework extends "chain-of-thought" reasoning to tables. In the chain-of-table approach, a dynamic planning process is employed, which utilizes LLMs to choose predefined operations. These operations are executed on the table at each step of the reasoning process.

The proposed chain-of-table framework demonstrates superior performance compared to both single-turn baseline methods for tabular reasoning and traditional textual chain-of-thought reasoning. This improved performance is validated on two table QA datasets, WikiTQ and FeTaQA, as well as TabFact, a dataset designed for table fact verification.

**Strengths:**

1. The proposed chain-of-table is simple and effective. This highlights the value of decomposing reasoning in tabular tasks, as opposed to employing single-step table reasoning methods.
2. The good performance underscores its effectiveness compared to baseline methods across multiple tabular reasoning datasets.

**Weaknesses:**

1. The proposed method is an extension of chain-of-thought to tabular data. Each reasoning step is constrained by predefined operations on the table. However, it raises questions about the adaptability of the chain-of-table framework to incorporate new operations or external knowledge, such as contextual information related to the table.
2. Chain-of-table requires a table, which could be large, in each reasoning step. This may significantly increase the computational cost. (See my questions below.)
3. While it is not necessarily a weakness, it would be beneficial to evaluate the proposed method with an open-sourced model (e.g. Llama-2) to understand whether the framework can be easily adopted by other models. One concern is that chain-of-table relies on LLMs' ability to comprehend the defined operations and reasoning chains, and it is uncertain whether other LLMs can seamlessly adapt to these requirements.

**Questions:**

1. In each dataset, could you clarify how many reasoning steps the chain-of-table method requires? Additionally, it would be helpful to understand the total number of input and output tokens required in comparison to the baselines.
2. For Figure 2, could you specify the number of examples utilized in each of its parts?

---

> ### Author Response · Authors · 2023-11-21
> **Response to reviewer eSFE (Part 1)**
>
> We would like to thank the reviewer eSFE for the insightful comments and suggestions. Please see response below:
>
> > it raises questions about the adaptability of the chain-of-table framework to incorporate new operations or external knowledge
>
> **[Response]** This is an excellent point. Regarding adaptability, Dater [A1] employs a dedicated and fixed procedure for table and question decomposition, which lacks compatibility with new operations. Binder [A2], while potentially compatible with new operations, is limited to those that work with code interpreters such as SQL or Python.
>
> We would like to highlight that Chain-of-Table represents the first initiative to overcome these limitations. It allows the adoption of an arbitrary number of new operations through flexible in-context demonstrations, without the need for operations to adhere to strict program executor formats.
>
> - [A1] Ye, Yunhu, et al. "Large Language Models are Versatile Decomposers: Decomposing Evidence and Questions for Table-based Reasoning." Proceedings of the 46th International ACM SIGIR Conference on Research and Development in Information Retrieval. 2023.
> - [A2] Cheng, Zhoujun, et al. "Binding Language Models in Symbolic Languages." The Eleventh International Conference on Learning Representations. 2023.

---

> ### Author Response · Authors · 2023-11-21
> **Response to reviewer eSFE (Part 2)**
>
> > While it is not necessarily a weakness, it would be beneficial to evaluate the proposed method with an open-sourced model.
>
> **[Response]** We follow the suggestion and conduct additional experiments using LLaMA (llama-2-13b-chat) on the TabFact benchmark. Our observations reveal that the proposed Chain-of-Table method achieves significant improvement over other recent baselines. This finding is in line with our submission, which uses PaLM 2 and GPT-3.5. The proposed method demonstrates consistent, model-agnostic enhancement for both closed and open models. We will include the results in the final version.
>
> |                | TabFact |
> | -------------- | ------- |
> | End-to-End QA  | 44.86   |
> | Fewshot QA     | 62.01   |
> | Dater          | 65.12   |
> | Chain-of-Table (ours) | 67.24   |

---

> ### Author Response · Authors · 2023-11-21
> **Response to reviewer eSFE (Part 3)**
>
> > In each dataset, could you clarify how many reasoning steps the chain-of-table method requires?
>
> **[Response]** In our framework, the selection of each operation is dynamically determined by the difficulty and complexity of the queries and their corresponding tabular inputs. Here we report the distribution of the required number of reasoning steps v.s. the number of samples.
>
> We observe that the majority of samples need 2 to 3 operations to generate the final output. Furthermore, TabFact typically requires more steps than WikiTQ. This is probably because TabFact is a fact-checking task, so the model naturally needs more steps to thoroughly verify all facts pertinent to the queries.
>
> | \# of operations | \# of samples in WikiTQ | \# of samples in TabFact |
> | ---------------- | ----------------------- | ------------------------ |
> | 0                | 42                      | 1                        |
> | 1                | 584                     | 4                        |
> | 2                | 2803                    | 547                      |
> | 3                | 738                     | 732                      |
> | 4                | 159                     | 517                      |
> | 5                | 18                      | 223                      |

---

> ### Author Response · Authors · 2023-11-21
> **Response to reviewer eSFE (Part 4)**
>
> > It would be helpful to understand the total number of input and output tokens required in comparison to the baselines.
>
> **[Response]** We follow the suggestion to compute the total number of input and output tokens required for Chain-of-Table compared to the top competing method, Dater, as shown in the table below. The 2nd column indicates the total tokens required, while the 3rd and 4th columns provide a detailed breakdown of the calculation process. The numbers in brackets in the 3rd and 4th columns represent the number of samples taken for self-consistency. We sample 20 examples from WikiTQ and use PaLM 2 as the backbone. The numbers below are computed on this subset.
>
> | Method         | Total input tokens per sample / Total output tokens per sample | Est. details of input tokens per sample                                 | Est. details of output tokens per sample                                           |
> | -------------- | ------------------------------------------------ | ----------------------------------------------------------------------- | ---------------------------------------------------------------------------------- |
> | Chain-of-Table (ours) | 12130 / 656                                      | DynamicPlan: 1654 (x 1 ~ 5); GenerateArgs: 445 ~ 2598; Query: 229           | DynamicPlan: 47; GenerateArgs: 47 ~ 86; Query: 3                                     |
> | Dater          | 33916 / 4360                                     | TableDecompose: 3774; Cloze: 927; Generate SQL: 1424 (x 20); Query: 735 | TableDecompose: 148 (x 20); Cloze: 30 (x 20); Generate SQL: 36 (x 20); Query: 4 (x 20) |
>
> The results show that the Chain-of-Table method requires significantly fewer input and output tokens compared to Dater, demonstrating its efficiency in token usage during the reasoning process.

---

> ### Author Response · Authors · 2023-11-21
> **Response to reviewer eSFE (Part 5)**
>
> > For Figure 2, could you specify the number of examples utilized in each of its parts?
>
> **[Response]** Figure 2 shows the prompt structure of DynamicPlan. We enumerate the three parts in the prompt along with the corresponding number of examples.
>
>
> | DynamicPlan Prompt Structure   | Explanation                                   | \# of Examples                           |
> | ------------------------------ | --------------------------------------------- | ---------------------------------------- |
> | Part 1 - Atomic Operation Demo | One example for each possible next operation. | 5                                        |
> | Part 2 - Operation Chain Demo  | A few examples of full operation chains       | 4 (in WikiTQ and TabFact); 3 (in FeTaQA) |
> | Part 3 - Input Sample          | The current example to be completed           | 1                                        |

---

### Official Review · Reviewer_4HBR · 2023-11-02

**Soundness:** 3 good
**Presentation:** 3 good
**Contribution:** 4 excellent
**Rating:** 6
**Confidence:** 3

**Summary:**

Chain-of-thought (CoT) prompting enables complex reasoning capabilities through intermediate reasoning steps. Still, these approaches incorporate the reasoning chain in a textual context and can't handle tabular data. The paper proposes a Chain-of-Table framework, where tabular data is explicitly used in the reasoning chain as a proxy for intermediate thoughts. The framework uses in-context examples to iteratively generate operations and update the table to represent a complex reasoning chain.

**Strengths:**

Recently, there has been a lot of interest in developing methods to improve the performance of LLMs on tabular data.
- The paper addresses one of the key stumbling blocks in improving the performance of LLMs on reasoning over tables.
- The paper is well-written and easy to follow.
- Empirical results show that the proposed approach outperforms the baselines on real-world datasets.

**Weaknesses:**

One of the issues with the paper is how they evaluate, especially the choice of benchmarks. I encourage the author to evaluate their approach over a diverse class of tasks, especially tasks like Table summarization, Column type annotation, Row augmentation, etc. In addition, I also encourage the authors to include open models too.

**Questions:**

Should the entire table fit inside the context length of the models? What happens if the tables are large? I would encourage the authors to evaluate their approach on larger tables.

In addition, a recent work that tries to modify CoT to leverage tabular structure [1] isn't discussed. I encourage the authors to distinguish with [1], further strengthening the submission.

Most existing LLMs often ignore the semantics of table structure, and it is beneficial to encode the structure in some form. Have the authors considered ways to incorporate the structure and semantics of the tables? As of now, the proposed techniques ignore them sans the operators defined and treat the table as yet another set of tokens.

[1] Jin Ziqi and Wei Lu. 2023. Tab-CoT: Zero-shot Tabular Chain of Thought. In Findings of the Association for Computational Linguistics: ACL 2023

---

> ### Author Response · Authors · 2023-11-21
> **Response to reviewer 4HBR (Part 1)**
>
> We would like to thank reviewer 4HBR for the insightful feedback. Please see response below:
>
> > I encourage the author to evaluate their approach over a diverse class of tasks
>
> **[Response]** We follow the suggestion and evaluate our approach on an additional table summarization task using the recently published QTSumm (https://github.com/yale-nlp/QTSumm) benchmark from EMNLP'23. The performance results are as follows.
>
> | Method         | BLEU  |
> | -------------- | ----- |
> | End2End        | 15.29 |
> | Few-shot       | 17.23 |
> | Chain-of-Table | 17.84 |
>
> Together with the table QA and table verification tasks in our submission, we observe that the proposed Chain-of-Table method is capable of reasoning over complex tabular data structures across 3 diverse tasks in total. Utilizing atomic operations, Chain-of-Table effectively decomposes the thought process and aggregates essential information, leading to consistently more reliable final answers. We will include the results in the final version.

---

> ### Author Response · Authors · 2023-11-21
> **Response to reviewer 4HBR (Part 2)**
>
> > In addition, I also encourage the authors to include open models too.
>
> **[Response]** We follow the suggestion and conduct additional experiments using LLaMA (llama-2-13b-chat) on the TabFact benchmark. Our observations reveal that the proposed Chain-of-Table method achieves significant improvement over other recent baselines. This finding is in line with our submission, which uses PaLM 2 and GPT-3.5. The proposed method demonstrates consistent, model-agnostic enhancement for both closed and open models. We will include the results in the final version.
>
> |                       | TabFact |
> | --------------------- | ------- |
> | End-to-End QA         | 44.86   |
> | Fewshot QA            | 62.01   |
> | Dater                 | 65.12   |
> | Chain-of-Table (ours) | 67.24   |

---

> ### Author Response · Authors · 2023-11-21
> **Response to reviewer 4HBR (Part 3)**
>
> >  Should the entire table fit inside the context length of the models? What happens if the tables are large?
>
> **[Response]** This is a great point! We observe that current tasks in table understanding do not surpass the context length limit of prevalent language model APIs. For instance, WikiTQ comprises 710/avg_token and 7615/max_token, TabFact comprises 464/avg_token and 1594/max_token, and FeTaQA comprises 397/avg_token and 2630/max_token.
>
> To further analyze the performance on larger tables, we categorize the input tables from WikiTQ into 3 groups based on token count: small (<2000 tokens), medium (2000 to 4000 tokens) and large (>4000 tokens). The results are as follows:
>
> | Table Size            | Small         | Medium        | Large          |
> | --------------------- | ------------- | ------------- | -------------- |
> | Dater                 | 62.50         | 42.34         | 34.62          |
> | Chain-of-Table (ours) | 68.13 (+5.63) | 52.25 (+9.91) | 44.87 (+10.25) |
>
> As anticipated, the performance decreases with larger input tables, as models are required to process and reason through longer contexts. Nevertheless, the performance of the proposed Chain-of-Table diminishes gracefully, achieving a significant 10+% improvement over the competing method. This demonstrates the efficacy of the reasoning chain in handling long tabular inputs.

---

> ### Author Response · Authors · 2023-11-21
> **Response to reviewer 4HBR (Part 4)**
>
> > In addition, a recent work that tries to modify CoT to leverage tabular structure [A1] isn't discussed. I encourage the authors to distinguish with [A1], further strengthening the submission.
>
> **[Response]** Thank you for pointing out Tab-CoT [A1]. We note that [A1] enhances CoT through a table-filling procedure, with a primary focus on text-based tasks where the input and output are in textual format. In contrast, our proposed Chain-of-Table method concentrates on table understanding. It performs reasoning directly on the input tables using atomic tabular operations. We will cite and discuss [A1] in the final version.
>
> - [A1] Jin Ziqi and Wei Lu. 2023. Tab-CoT: Zero-shot Tabular Chain of Thought. In Findings of the Association for Computational Linguistics: ACL 2023

---

> ### Author Response · Authors · 2023-11-21
> **Response to reviewer 4HBR (Part 5)**
>
> > Have the authors considered ways to incorporate the structure and semantics of the tables?
>
> **[Response]** In alignment with prior studies [A2, A3, A4] and the baseline methods [A5, A6], we adopt PIPE encoding for the proposed method. This decouples the performance gains of the proposed tabular CoT with atomic operations from the influence of various table formatting choices.
>
> To further understand the impact of different semantic encodings on table understanding performance, we conduct additional experiments using 3 additional table representations: HTML, TSV, and Markdown. For these experiments, we use End-to-End QA on WikiTQ with PaLM 2 as a running example. The results are as follows:
>
>
> | WikiTQ  | PIPE  | HTML  | TSV   | Markdown |
> | ------- | ----- | ----- | ----- | -------- |
> | Accuracy | **0.606** | 0.561 | 0.581 | 0.58     |
>
> These findings are in line with reviewer 4HBR's insight that different semantic encodings indeed lead to different outcomes. Notably, the PIPE format adopted in our study yields the highest performance among the four encodings tested. Examples of PIPE-encoded tables are included in the Appendix.
>
> - [A2] Liu, Qian, et al. "From zero to hero: Examining the power of symbolic tasks in instruction tuning." arXiv preprint arXiv:2304.07995 (2023).
> - [A3] Liu, Qian, et al. "TAPEX: Table Pre-training via Learning a Neural SQL Executor." International Conference on Learning Representations. 2021.
> - [A4] Jiang, Zhengbao, et al. "OmniTab: Pretraining with Natural and Synthetic Data for Few-shot Table-based Question Answering." Proceedings of the 2022 Conference of the North American Chapter of the Association for Computational Linguistics: Human Language Technologies. 2022.
> - [A5] Ye, Yunhu, et al. "Large Language Models are Versatile Decomposers: Decomposing Evidence and Questions for Table-based Reasoning." Proceedings of the 46th International ACM SIGIR Conference on Research and Development in Information Retrieval. 2023.
> - [A6] Cheng, Zhoujun, et al. "Binding Language Models in Symbolic Languages." The Eleventh International Conference on Learning Representations. 2023.

---

> > ### Comment · Reviewer_4HBR · 2023-11-23
> >
> > Thanks for the detailed response, I will keep my accept recommendation.

---

### Meta-Review · Area_Chair_PUSV · 2023-12-22

**Metareview:**

This is a decently strong paper. It works on Table-Reasoning problems using LLMs by developing a reasoning chain, in which at each step of iteration, one of five reasoning steps are proposed by the LLM, which may result in modification of the table (e.g., by adding columns, or aggregating information) and eventually the final answer is obtained.

As an idea, it is a great idea, though not original. as other previous work (as cited in the paper) have started exploring the high level idea. The paper builds them further -- e.g., proposes five types of reasoning steps where previous work (e.g., BINDER) only does two types. One other characteristic is the iterative nature where a new table may get created at each step of iteration. The results as shown in the paper are extremely strong and strongly outperform existing methods.

**Justification For Why Not Higher Score:**

There is some incrementality in terms of ideas compared to BINDER work. Better exposition (both experimental and algorithmic) could have brought out the differences better. That said, the paper is not _that_ groundbreaking to suggest a spotlight/oral.

**Justification For Why Not Lower Score:**

On further thought, there are two issues to be considered here. One, there are many similarities between BINDER paper and CHAIN OF TABLES paper. There are some differences also (e.g., 5 operations vs 2 API calls). So, exposition of clear differences (And without mistakes) will help. However, the paper also shows that their work strongly outperforms both BINDER and its successor DATER.

This should be enough of an evidence that the new work is salient. And the authors can simply explain the differences with BINDER paper better in the revision. But, in the process of analyzing this more, I noticed that the results published in this paper are weaker than those reported in DATER paper. The authors themselves acknowledge it in Footnote 3 that because Codex is not available they could not experiment in that setting. I find that unfortunate from the purpose of science, but satisfactory from the point of view of the authors did what they could have.

So, all in all, I feel that the paper is decently good. Good technical ideas and excellent results (but with weaker than previous LLMs). Some issues are still open. Why is your method that much better than BINDER -- what is the source of power of your work? Can you do controlled experiments to show the components that are adding strong results? I trust that the authors will answer these questions with their best effort in the next version.

---

### Decision · Program_Chairs · 2024-01-16

Accept (poster)